# Association between depression during pregnancy and preterm birth: Results from population cohorts and mouse experimental models

Siguo Chen[1], Guanghong Yan[2], Xinzi Xie[2], Qihan Wang[2], Jie Zhong[2], Qian Wang[2], Jinman Zhang[3], Hongying Li[4], Dingyun You[2]*

1 Yunnan Cancer Hospital, The Thrid Affiliated Hospital of Kunming Medical University, Department of Thoracis Surgery, Kunming, Yunnan, China, 2 School of Public Health, Kunming Medical University, Kunming, Yunna, China, 3 Department of Medical Genetics, First People's Hospital of Yunnan Province, Kunming, Yunnan, China, 4 Department of Obstetrics, Beicheng Hospital, First People's Hospital of Qujing, Yunnan, China

☯ This author contributed equally to this work and are considered co-first authors.
* youdingyun@kmmu.edu.cn

## Abstract

### Background

Depression is a prevalent psychological challenge during pregnancy, with established links to adverse outcomes like preterm birth (PTB) globally. However, epidemiological data from China's multiethnic regions are scarce, and experimental evidence supporting a causal relationship remains limited. This study aimed to investigate the association between prenatal depressive symptoms and the risk of PTB in a cohort from Yunnan, China, and to provide supportive evidence using a mouse model of depression.

### Methods

We recruited 1,466 women during their first-trimester routine visits at Qujing Hospital. Depressive symptoms were assessed using the Chinese version of the Edinburgh Postnatal Depression Scale (EPDS), a screening tool, with a score ≥12 indicating elevated symptoms suggestive of depression. PTB was defined as delivery before 37 gestational weeks, confirmed by ultrasound. In parallel, a mouse model of depression was established using Chronic Unpredictable Mild Stress (CUMS) for 6 weeks prior to mating. PTB in mice was defined as delivery before 19 days of gestation.

### Results

In the cohort study, the incidence of PTB was significantly higher in women with prenatal depressive symptoms compared to those without (8.43% vs. 3.83%, P < 0.001).

**Data availability statement:** All relevant data are within the paper and Supporting information files.

**Funding:** This study was financially supported by the National Natural Science Foundation of China (https://www.nsfc.gov.cn) in the form of grants received by DY (81960592 and 82073569) and JZ (82260645). This study was also financially supported by the Applied Basic Research Key Project of Yunnan in the form of a grant (202101AS070040) received by DY. This study was also financially supported by the Special Project for Selection of High-level Scientific and Technological Talents and Innovation Team-Reserve Talent Program for Young and Middle-aged Academic and Technical Leaders in the form of a grant (202005AC160023) received by DY. This study was also financially supported by the Basic Research of Yunnan Province-Excellent Youth Project in the form of a grant (202001AW070021) received by DY. This study was also financially supported by the Yunnan Province Joint Special Project of Kunming Medical University in the form of a grant (202101AY070001) received by DY. This study was also financially supported by the Doctoral Research Fund Project of the First Affiliated Hospital of Kunming Medical University in the form of a grant (2022BS013) received by DY. The funders had no role in study design, data collection and analysis, decision to publish, or preparation of the manuscript.

**Abbreviations:** ARRIVE: Animal Research: Reporting of In Vivo Experiments; CUMS, Chronic Unpredictable Mild Stress; EPDS, Edinburgh Postnatal Depression Scale; FST, Forced Swim Test; NIH, National Institutes of Health; OFT, Open Field Test; PTB, Preterm Birth; SPF, Specific Pathogen Free; SPT, Sucrose Preference Test; TST, Tail Suspension Test

The association remained significant after adjusting for sociodemographic and clinical confounders, with an adjusted risk ratio (aRR) of 2.19 (95% CI: 1.32–3.63). This association showed a significant dose-response pattern (P for trend = 0.03), with the risk being highest for women with moderate depressive symptoms (aRR = 2.44, 95% CI: 1.30–4.58). In the animal experiments, PTB did not occur in the control mice, whereas 40% of the mice exposed to CUMS (depression model group) delivered prematurely.

## Conclusion

This study demonstrates a significant association between prenatal depressive symptoms and an increased risk of preterm birth in a Chinese multiethnic cohort. Experimental findings from a mouse model further suggest a potential contributory role of depression to PTB. These results underscore the importance of screening for and addressing maternal mental health during pregnancy. Future research should focus on underlying mechanisms and intervention strategies to mitigate this risk.

## Introduction

Depression is a common mental disorder during pregnancy [1]. Changes in a woman's physical and psychological status during pregnancy increase the risk of depression when influenced by factors such as social and cultural factors [2–4]. The global incidence ranges from 15% (European cohorts, 2015–2020) [5] to 65% (rural Pakistan, 2018–2020) [6], with low-income countries showing steeper increases (annual growth rate = 2.3% vs.0.8% in high-income nations). The prevalence of depression during pregnancy is significantly higher in low- and middle-income countries than in high-income countries, and the prevalence of depression during pregnancy is increasing in low-income countries [7]. In recent years, the occurrence of depression during pregnancy in China has shown an upward trend [8]. A comparison of existing evidence has found that the prevalence of depression during pregnancy in China has increased from less than 10% around 2008 to over 20% in more recent studies [9,10].

Preterm birth (PTB), a primary cause of neonatal morbidity and mortality, is a syndrome influenced by multiple risk factors, including infection/inflammation, physiological stress, and socio-economic determinants [11,12]. Among these, prenatal depression can affect fetal growth and is a recognized risk factor for adverse outcomes such as PTB [13,14]. A major objective of this study is to elucidate the specific biological and stress-mediated pathways through which prenatal depression contributes to PTB, which remains incompletely understood [15–17].The strength of the association between prenatal depression and PTB appears to vary between ethnic groups, with studies suggesting a stronger association in some Asian populations [7,18–23]. We hypothesize that this variation may be linked to culturally specific psychosocial stressors, varying social support structures, and genetic factors. Yunnan, as a multiethnic region, provides a critical context to investigate this [24–26]. It is characterized by significant demographic diversity, including unique patterns of minority population

distribution and migration that distinguish it from many other Chinese provinces. For instance, studies show that Yunnan's unique ethnic minorities, who were once highly concentrated, are now experiencing increased out-migration and inter-provincial dispersal. This dynamic, coupled with diverse cultural norms around pregnancy, may modulate depression risk and its physiological impacts. As the association between prenatal depression and PTB has not been specifically reported in this unique regional context, it remains to be determined whether the effect differs from other regions.

Animal models are indispensable for validating clinical observations and investigating the causal mechanisms underlying PTB, with established models—including those for infection, inflammation, and hormone induction—effectively replicating aspects of human parturition. Furthermore, experimental studies in animals provide direct evidence that stress during gestation has profound developmental consequences. Research demonstrates that prenatal stress can disrupt fetal neuroplasticity and program long-term changes in brain function, increasing susceptibility to mental illness later in life. Supporting this, a recent clinical study found that maternal anxiety during pregnancy is associated with measurable alterations in the fetal brain, which may explain subsequent neurodevelopmental problems in children [25–28]. Therefore, employing a Chronic Unpredictable Mild Stress (CUMS) model in mice allows us to test the hypothesis that depression-like behaviors can directly contribute to PTB and to explore the underlying pathophysiological pathways.

Based on the established link between stress and adverse outcomes, we hypothesized that:Prenatal depressive symptoms are significantly associated with an increased risk of PTB in the multiethnic population of Yunnan, China, and inducing depression-like behaviors in a mouse model would result in a higher incidence of PTB, supporting a potential causal role.

## Materials and methods

### Study design

This study used a combination of prospective cohort design and experimental animal models and a quantitative study design that did not include qualitative methods because the primary objective was to quantify the association between depressive symptoms and risk of preterm birth, rather than to explore subjective experience.

### Study population

Pregnant women receiving prenatal care at Kunming Medical University Affiliated Qujing Hospital from August 2020 to March 2022 constituted the study subjects. Inclusion criteria comprised singleton pregnancy, proficiency in reading comprehension and communication, voluntary participation in the survey, and adherence to the premature labor diagnosis. Exclusion criteria encompassed severe medical and surgical conditions (e.g., gestational diabetes, gestational hypertension, preeclampsia, congenital heart disease, psychiatric disorders), use of antidepressants and other psychotropic drugs, and history of stillbirth, birth defects, miscarriage, or incomplete information. The sample size was calculated a priori using power analysis. With an assumed PTB rate of 5% in the non-exposed group and 10% in the exposed (depressive symptoms) group, an alpha of 0.05, and a power of 80%, a minimum of 1,366 participants was required. Accounting for a potential 10% loss to follow-up, we aimed to recruit at least 1,500 pregnant women.

### A questionnaire and content

A customized questionnaire was employed to gather demographic details of pregnant women and information pertaining to depression during pregnancy. Demographic data encompassed age, ethnicity, education level, occupation, marital status, annual household income, place of residence, and pregnancy-related health status.

### Screening tools for depression during pregnancy

Assessment of depressive symptoms was conducted using the EPDS, a widely used screening tool for postpartum depression, which is equally applicable for screening depressive symptoms during pregnancy. It is crucial to note that the

EPDS identifies symptoms suggestive of depression but is not a diagnostic tool for clinical depression. The scale gauged the severity of depressive symptoms over the past 7 days, comprising 10 items graded as "never," "occasionally," "often," and "always," corresponding to "0 points," "1 point," "2 points," and "3 points," respectively. The total score ranged from 0 to 30 points, with higher scores indicating more severe depressive symptoms. Participants were categorized into two groups based on their EPDS scores: women with elevated depressive symptoms (EPDS ≥ 12) and women without elevated depressive symptoms (EPDS < 12) using 12 points as the cutoff value. Scores of 12–13, 14–18, and 19–30 indicated mild, moderate, and major depressive symptoms, respectively [29–32]. To ensure data quality, all questionnaires were administered by trained staff. Electronic data entry was performed with built-in validation checks, and a random sample of 10% of the paper records was double-checked for accuracy.

## Establishment of a murine model for depression

The CUMS model is currently the most widely used, reliable, and effective rodent model for depression [33]. By employing various environmental and physical stimuli, animals are subjected to long-term exposure under unpredictable conditions, effectively replicating the diverse psychological pressures encountered in human daily life. This depression model extends over three months and is widely utilized in most experimental studies.Selection and Grouping of Mice A 6-week-old SPF-level C57BL/6 female mouse weighing 18.0–21.0 g was selected from the Department of Laboratory Animal Science, Kunming Medical University, and assigned an experimental unit use license number. The experimental animals were housed in a clean animal room with a maintained room temperature of 25 ± 1°C, humidity of 65 ± 5%, and a 12-hour cyclic alternating illumination method for the circadian rhythm (light time from 7 am to 19 pm). After one week of adaptive rearing, all mice underwent open field experiments. Mice with poor spontaneous movement were eliminated based on the open field experiment score, the sample size in the CUMS group was larger than in the control group to account for anticipated individual variability in response to the stress protocol and to ensure an adequate number of animals met the depression criteria for subsequent pregnancy analysis, and the remaining 100 mice were randomly divided into a control group (n = 20) and a CUMS depression model group (n = 80) according to body weight. The control group was housed normally in one cage every 5 animals, while the CUMS group was individually housed.Induction of Depressive Stimulation in Mice The CUMS depression model group was subjected to various stimuli, including physical stressors: binding for 6 hours, abstaining from drinking for 24 hours, fasting for 24 hours, cold water bath for 3 minutes, hot bath for 3 minutes, and tail clip for 2 minutes; and environmental stressors: tilting for 12 hours, wet litter overnight, day and night alternation for 8 hours, strobe for 8 hours, exposure to strange abnormal objects overnight, and long day for 24 hours. To induce the formation of depression, 2–3 stimuli were given every day, with the same stimulus not appearing more than 2 times a week, for a total of 6 weeks. Observation of Mice The general condition of all mice was observed daily, including hair color, activity, mental state, body posture, struggle response to grasping, stress response, and urine. The two groups of mice were weighed before modeling, on the 1st, 2nd, 3rd, 4th, 5th, and 6th weekends, and daily after conception, and the changes in mouse body weight at different time points were recorded.

## Behavioral tests

**Sugar preference test (SPT).** The SPT was utilized to assess hedonic deficiency in mice, with a decrease in sugar water preference indicative of anhedonia [34]. Three days prior to commencing the experiment, the mice underwent training to acclimate to the sugar water test. Each mouse was group-housed in a single cage, with two water bottles provided simultaneously within each cage. Initially, both bottles contained 1% sucrose water. Subsequently, one bottle contained sucrose water, while the other contained distilled water, with the positioning of the bottles rotated every 4 hours to prevent bias. On the third day, food and water were withheld. At 9 am on the fourth day, each mouse was simultaneously given prequantified sucrose water and distilled water, and after 24 hours, all the water bottles were

weighed. Throughout the modeling phase, measurements were taken on the 2nd, 4th, and 6th days according to the described protocol, and the sugar water preference rate was calculated as a percentage of total water consumption (1% sugar water consumption/total water consumption×100%).

**Open field test (OFT).** The OFT can detect spontaneous locomotion and exploratory behavior in mice, and the effect is significant [35,36]. In this experiment, an open field experiment was performed before the modeling and on the weekends of the 2nd, 4th, and 6th weeks of the molding, and the mice were placed in the laboratory for 60 minutes before the start of the experiment to adapt to the laboratory environment. A single mouse was placed in the open field test chamber during observation, and the open field data were analyzed after the overall experiment.

**Forced swimming test (FST).** The FST is a rodent behavior test commonly used to evaluate depressive-like behavior in model animals with good confidence and predictive validity [37,38]. In this experiment, a forced swimming experiment was performed before the modeling, and one forced swimming experiment was performed on the weekends of the 2nd, 4th, and 6th days of the modeling. After placing the mouse in a cylindrical glass filled with water for 6 min and acclimatizing for the first 2 min, the remaining time for the mouse was recorded for the next 4 min, and the mouse was judged to be immobile when it floated in an upright position with its head exposed to the surface of the water. The water was changed after each test, and the laboratory environment was kept quiet during the experiment.

**Tail suspension test (TST).** The TST is generally used for screening antidepressants and for behavioral evaluation in animal models [39]. The whole process of the experiment usually lasts 6 minutes, and the duration of immobility is generally evaluated as a state of despair in mice. This experiment was performed once before molding and once on the weekends of the 2nd, 4th, and 6th weeks of molding. With a homemade partition 50 cm from the ground, one side is open, the mouse is easy to observe, and the rest of the side is closed. To avoid interference from the external environment, the mouse tail is glued to the crossbar placed on the partition beforehand, and the amount of time it does not move is recorded. The mice were allowed to acclimatize to the environment for the first 2 min, and the immobility time was recorded for the last 4 min. The laboratory environment was kept quiet during the experiment.

**Animal handling and welfare considerations.** Throughout the study, we were committed to minimizing stress beyond the experimental protocols. While specific behavioral tests (TST) inherently required tail handling as part of the procedure, all other routine animal management activities—including daily health checks, cage transfers, and animal identification—were performed using low-stress techniques. Wherever feasible, handling tunnels or cupping methods were employed instead of direct tail grasping. These measures aimed to improve animal welfare and reduce baseline stress that could confound behavioral assessments.

**Mouse cage conception and pregnancy observation.** Successful depression induction required ≥30% decrease in sucrose preference vs. controls (P<0.01), ≥40% increase in immobility time (FST/TST), and consistent weight divergence (>15%) by week 3. These thresholds align with published CUMS standards. Nine-week-old SPF-level C57BL/6 male mice and mature depressed models were carefully selected to establish successful mating pairs for mice and controls, aiming to enhance the pregnancy rate. Mating was conducted at a male-to-female ratio of 1:2, with male mice introduced into the cages at 18:00 on the day of mating and removed at 8:00 the following day. Subsequently, the female mice were directly observed for the presence of a vaginal plug using forceps, indicating successful mating. Mating usually occurred at night, and the identification of the vaginal plug marked the suspected pregnancy day. To prevent oversight and false pregnancy, the body weight of pregnant mice was monitored daily. Suspected pregnancies were confirmed if the body weight of the mice increased by more than 2 g within 10 days, and these mice were then individually housed and given special care. Nonpregnant mice underwent a second mating cycle using the same method. From day 17, the mice were observed every 12 hours to detect the onset of labor. The normal gestation period for mice is 19–21 days, with each 12-hour interval recorded as 0.5 days. PTB was noted for gestation periods of less than 19 days. The pregnancy rate, pregnancy loss (including biochemical pregnancies, fetal arrest, or miscarriages), premature birth rate, and other indicators were calculated, along with the number of litters, gestational weight gain, and neonatal mouse body weight. At the conclusion

of the study, mice were euthanized via cervical dislocation under isoflurane anesthesia. No invasive procedures requiring analgesia were performed during behavioral tests. Efforts to minimize suffering included daily health monitoring, environmental enrichment (nesting materials), and immediate intervention for signs of severe distress (none observed). Following the 6-week CUMS paradigm, individual mouse model validation was performed. Mice that did not meet the pre-defined criteria for depression-like behavior (i.e., a ≥ 30% decrease in sucrose preference and a ≥ 40% increase in immobility time in the FST or TST compared to the control group average) were excluded from subsequent mating and pregnancy analyses to ensure the validity of the depression model. Based on these criteria, 20 mice from the original CUMS group (n = 80) were excluded, resulting in 60 validated depressed mice proceeding to the mating phase.

**Ethical review and informed consent.** All parts of the study involving human participants were conducted in accordance with the Declaration of Helsinki, and informed consent was obtained from the participants. This study was approved by the Medical Ethics Committee of Kunming Medical University and met the relevant ethical standards (KMMU2020MEC056). All participants voluntarily participated in the study and signed an informed consent form, and patient information was kept confidential. This study followed the STROBE reporting guidelines. All animal testing procedures for this study were conducted in accordance with the "ARRIVE" guidelines and were approved by the Ethics Review Board of Kunming Medical University (kmmu20211287) and followed the NIH Guide for the Care and Use of Laboratory Animals. Veterinarians monitored animals twice daily using a 12-point distress scale.

## Statistical analysis

Statistical analysis of the population cohort and animal experiments was performed using R language software (version 4.2.0) and GraphPad Prism (version 9.0.0). The analyses proceeded as follows: First, we described the demographic characteristics of the cohort. Second, we assessed the association between prenatal depressive symptoms (EPDS ≥12) and PTB using Cox proportional hazards models, which included Model 1: no adjustment for confounding factors; Model 2: adjusted for age, nationality, residence, marital status, education level, occupation status, annual family income, and pregnancy. Model 3: Adjust for all above plus pre-pregnancy BMI and parity. Third, a pre-planned analysis examined the dose-response relationship by categorizing depressive symptoms into mild, moderate, and severe levels. Fourth, we conducted pre-specified subgroup analyses to explore effect modification by age, trimester of enrollment, socioeconomic status, education, residence, occupation, and marital status. If sufficient data are available from the cohort, pre-specified subgroup analyses will be conducted to examine the association between depressive symptoms and PTB across strata defined by maternal age (<25, 25–34, ≥ 35 years), socioeconomic status (low/high), and place of residence (urban/rural). A Bonferroni correction was applied to account for multiple comparisons in the subgroup analyses. The trimester of enrollment was determined based on the gestational age at the time of the first-trimester visit when the demographic questionnaire and EPDS were administered. All tests were two-tailed with α = 0.05, and P < 0.05 was considered statistically significant.

## Results and discussion

### Human cohort results

**Participant characteristics.** A total of 1,466 pregnant women were included in the final analysis (Fig 1). Based on EPDS scores at the first-trimester visit, 320 (21.83%) were classified as having elevated depressive symptoms. The demographic and clinical characteristics of the participants, stratified by depressive symptom status, are presented in Table 1. The distribution of participants by age, ethnicity, marital status, occupation, and gestational week at enrollment did not differ significantly between groups (P > 0.05). Differences were observed in residence, education level, and annual household income (P < 0.05).

**Association between depressive symptoms and preterm birth and the dose-response relationship.** Depressed patients were significantly more likely to develop PTB during pregnancy than

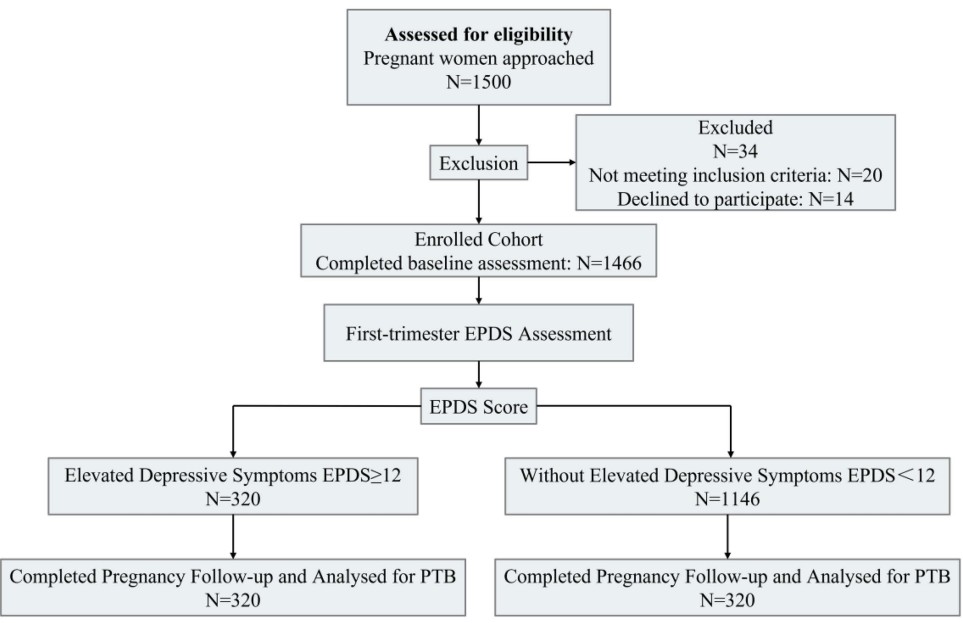

**Fig 1. Flowchart of participant selection in the prospective cohort study.**

non-depressed patients (S1Table). Different depression scores also have different effects on pregnancy outcomes (Fig 2). The incidence of PTB was significantly higher in women with elevated depressive symptoms (EPDS ≥12) than in those without (8.43% [27/320] vs. 3.83% [44/1146], P < 0.001). Table 2 presents the results of the association and dose-response analysis across three sequential models. In the unadjusted model (Model 1), elevated depressive symptoms were associated with a 2.31-fold increased risk of PTB (RR = 2.31, 95% CI: 1.41–3.79). This association remained significant after adjustment for sociodemographic and clinical factors in Model 2 (aRR = 2.19, 95% CI: 1.32–3.63) and further persisted in the fully adjusted model (Model 3) that included pre-pregnancy BMI and parity (aRR = 2.08, 95% CI: 1.25–3.47).

A significant dose-response relationship was observed between the severity of depressive symptoms and the risk of PTB (P for trend = 0.04 in Model 3). Specifically, in the fully adjusted model (Model 3), compared to the reference group (EPDS <12), moderate depressive symptoms (EPDS 14–18) were associated with a 2.35-fold increased risk of PTB (aRR = 2.35, 95% CI: 1.24–4.44). The association for mild depressive symptoms (EPDS 12–13) was not statistically significant (aRR = 1.88, 95% CI: 0.91–3.91). The analysis for the severe depressive symptoms (EPDS ≥19) subgroup was limited by a small sample size, resulting in a non-statistically significant association with wide confidence intervals (aRR = 1.61, 95% CI: 0.36–7.19).

**Subgroup analysis.** Pre-specified subgroup analyses were performed to assess the consistency of the association between depressive symptoms and PTB across different population strata (Fig 3). The increased risk of PTB associated with depressive symptoms was observed consistently across most subgroups. The point estimates were highest for women who were unemployed, resided in urban areas, or had a college/bachelor's degree or above.

In animal experiments, depression during pregnancy has been shown to have pronounced effects on adverse pregnancy outcomes. To further elucidate the causal relationship between pregnancy depression and adverse pregnancy outcomes observed in the human population, this study conducted animal experiments, establishing a mouse depression model and observing various indicators. The results revealed that depression significantly impacted the general condition, pregnancy rate, and PTB rate of mice.

**Table 1. Statistical description of demographic characteristics.**

| Variable | All (n = 1466) | Depression | | P value |
| --- | --- | --- | --- | --- |
| | | No (n = 1146) | Yes (n = 320) | |
| Age (years) | | | | 0.081 |
| <25 | 199 (13.57) | 148 (74.37) | 51 (25.63) | |
| 25~34 | 1125 (76.74) | 878 (78.04) | 247 (21.96) | |
| ≥35 | 142 (9.69) | 120 (84.51) | 22 (15.49) | |
| Minority | | | | 0.724 |
| Yes | 119 (8.12) | 91 (76.47) | 28 (23.53) | |
| No | 1347 (91.88) | 1055 (78.32) | 292 (21.68) | |
| Residence | | | | 0.001 |
| Town | 847 (57.78) | 688 (81.23) | 159 (18.77) | |
| Rural | 619 (42.22) | 458 (73.99) | 161 (26.01) | |
| Marital status | | | | 0.928 |
| Married/cohabitating | 1426 (97.27) | 1114 (78.12) | 312 (21.88) | |
| Single/divorced | 40 (2.73) | 32 (80.00) | 8 (20.00) | |
| Education level | | | | 0.008 |
| At least primary school level | 216 (14.73) | 154 (71.30) | 62 (28.70) | |
| At least secondary school level | 368 (25.10) | 281 (76.36) | 87 (23.64) | |
| Advanced level and tertiary level | 882 (60.16) | 711 (80.61) | 171 (19.39) | |
| Occupation | | | | 0.113 |
| Employed | 1231 (83.97) | 972 (78.96) | 259 (21.04) | |
| Not employed | 235 (16.03) | 174 (74.04) | 61 (25.96) | |
| Average household income | | | | 0.023 |
| ≤$16,500 | 1239 (84.52) | 955 (77.08) | 284 (22.92) | |
| >$16,500 | 227 (15.48) | 191 (84.14) | 36 (15.86) | |
| Pregnant (weeks) | | | | 0.814 |
| Early pregnancy (<14) | 186 (12.69) | 147 (79.03) | 39 (20.97) | |
| Mid-pregnancy (14~28) | 352 (24.01) | 271 (76.99) | 81 (23.01) | |
| Late pregnancy (≥28) | 928 (63.30) | 728 (78.45) | 200 (21.55) | |
| Birth weight | 3194.25 (421.27) | 3194.29 (413.75) | 3194.10 (448.10) | 0.995 |
| Depression score | 8.62 (4.32) | 6.90 (2.87) | 14.81 (2.68) | <0.001 |

## Mouse experimental results

**Successful establishment of the depression model.** Of the 80 mice initially subjected to CUMS, 60 (75%) met the predefined behavioral criteria for a successful depression model and were included in subsequent mating and pregnancy analyses. A total of 60 female mice were subjected to CUMS to induce depression-like behaviors. Beginning from the first week of modeling, the body weight of the CUMS group was significantly lower than that of the control group (P<0.05, Fig 4A). Behavioral tests conducted at weeks 2, 4, and 6 of the modeling period revealed the following in the CUMS group compared to the control group:In the OFT, the motion trajectory was altered (Fig 4B), with a significant decrease in the central grid dwell time (Fig 4C), total distance traveled (Fig 4D), and the percentage of central grid distance to total distance (Fig 4E).

In the SPT, the sugar water preference rate was significantly decreased (Fig 4F).In the TST and FST, the immobility time was significantly increased (Fig 4G, 4H). All comparisons yielded P<0.05. The trends of these behavioral changes are detailed in S2 Table.

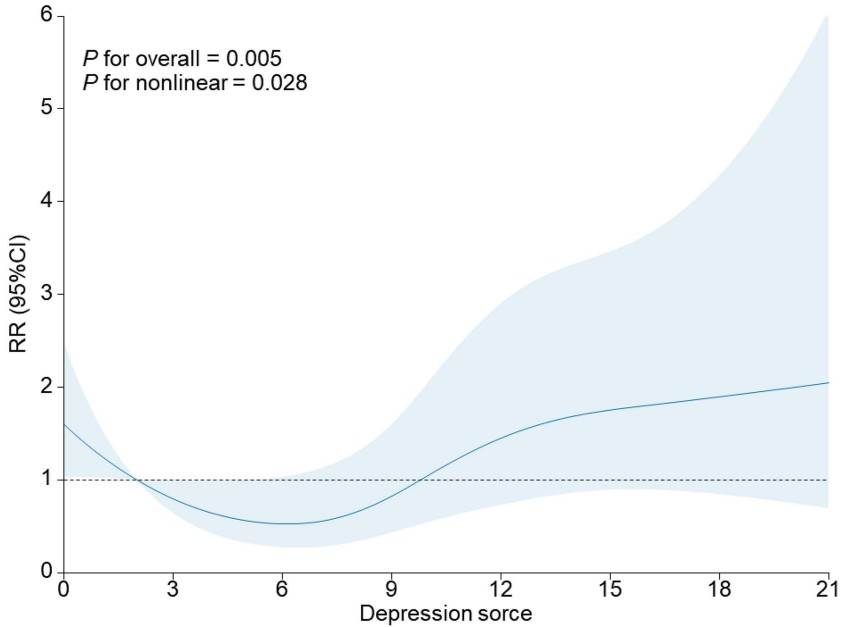

**Fig 2. Association of predicted depression scores with Premature delivery rate.**

**Table 2. Association of depression degree with PTB.**

| Pregnant depression | Model 1 | | Model 2 | | Model 3 | |
|---|---|---|---|---|---|---|
| | RR(95%CI) | P value | RR(95%CI) | P value | RR(95%CI) | P value |
| Normal | Ref | | Ref | | Ref | |
| Depressive | 2.31 (1.41, 3.79) | < 0.001 | 2.19 (1.32, 3.63) | 0.002 | 2.08(1.25,3.47) | 0.005 |
| Mild depressive | 2.14 (1.05, 4.37) | 0.036 | 2.01 (0.97, 4.15) | 0.061 | 1.88(0.91,3.91) | 0.089 |
| Moderate depressive | 2.54 (1.38, 4.67) | 0.003 | 2.44 (1.3, 4.58) | 0.005 | 2.35(1.24,4.44) | 0.009 |
| Severe depressive | 1.79 (0.41, 7.75) | 0.437 | 1.67 (0.38, 7.36) | 0.495 | 1.61(0.36,7.19) | 0.534 |
| P-trend | 0.015 | | 0.03 | | 0.04 | |

Model 1: no adjustment for confounding factors.

Model 2: adjust age, nationality, residence, marital status, education level, occupation status, annual family income, pregnancy.

Model 3:adjust for all above + pre-pregnancy BMI and parity.

**Pregnancy and preterm birth outcomes.** Following the mating procedure, the pregnancy rate in the CUMS group was 53.33% (32/60), which was lower than the 80.00% (16/20) observed in the control group (S3 Table).Among the successfully conceived mice, the incidence of PTB was 40.00% (12/30) in the CUMS group. In contrast, no PTB cases (0/15) occurred in the control group. This difference in pregnancy outcomes between the two groups was statistically significant (P<0.05, Table 3).

**Litter size and neonatal weight.** No significant differences were observed between the CUMS and control groups in maternal weight gain during pregnancy (S4 Table), litter size (S1 Fig), or the body weight of neonatal mice (P>0.05).

## Principal findings

This prospective cohort study demonstrates a significant association between elevated depressive symptoms in early pregnancy and an increased risk of PTB among a multiethnic population in Yunnan, China. This association followed a

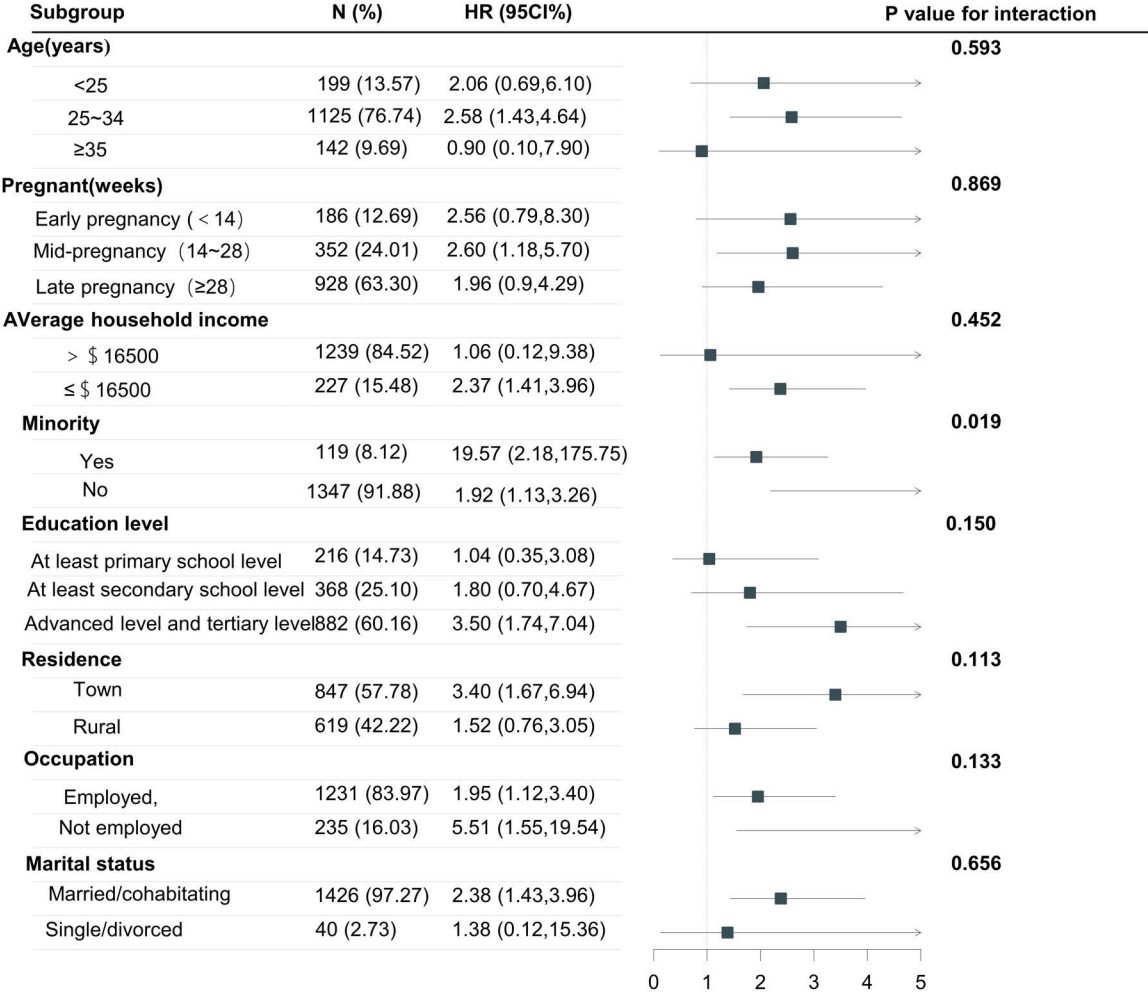

| Subgroup | N (%) | HR (95CI%) | P value for interaction |
|---|---|---|---|
| **Age(years)** | | | 0.593 |
| <25 | 199 (13.57) | 2.06 (0.69,6.10) | |
| 25~34 | 1125 (76.74) | 2.58 (1.43,4.64) | |
| ≥35 | 142 (9.69) | 0.90 (0.10,7.90) | |
| **Pregnant(weeks)** | | | 0.869 |
| Early pregnancy (< 14) | 186 (12.69) | 2.56 (0.79,8.30) | |
| Mid-pregnancy (14~28) | 352 (24.01) | 2.60 (1.18,5.70) | |
| Late pregnancy (≥28) | 928 (63.30) | 1.96 (0.9,4.29) | |
| **AVerage household income** | | | 0.452 |
| > $ 16500 | 1239 (84.52) | 1.06 (0.12,9.38) | |
| ≤ $ 16500 | 227 (15.48) | 2.37 (1.41,3.96) | |
| **Minority** | | | 0.019 |
| Yes | 119 (8.12) | 19.57 (2.18,175.75) | |
| No | 1347 (91.88) | 1.92 (1.13,3.26) | |
| **Education level** | | | 0.150 |
| At least primary school level | 216 (14.73) | 1.04 (0.35,3.08) | |
| At least secondary school level | 368 (25.10) | 1.80 (0.70,4.67) | |
| Advanced level and tertiary level | 882 (60.16) | 3.50 (1.74,7.04) | |
| **Residence** | | | 0.113 |
| Town | 847 (57.78) | 3.40 (1.67,6.94) | |
| Rural | 619 (42.22) | 1.52 (0.76,3.05) | |
| **Occupation** | | | 0.133 |
| Employed, | 1231 (83.97) | 1.95 (1.12,3.40) | |
| Not employed | 235 (16.03) | 5.51 (1.55,19.54) | |
| **Marital status** | | | 0.656 |
| Married/cohabitating | 1426 (97.27) | 2.38 (1.43,3.96) | |
| Single/divorced | 40 (2.73) | 1.38 (0.12,15.36) | |

**Fig 3. Subgroup analysis of depression during pregnancy.**

dose-response pattern for mild to moderate symptoms and was robust to adjustment for multiple confounders. Supporting these observational findings, our experimental study in a mouse model of depression showed that depression-like behaviors could directly lead to a significantly higher incidence of PTB. A more detailed characterization of our cohort with depressive symptoms revealed that for most, the onset occurred during pregnancy, and low social support was a prevalent psychosocial factor. However, the lack of detailed clinical history for all participants is a limitation that future studies should address.

## Comparison with previous studies

Our finding of a 2.19-fold increased risk (Model 2) is consistent with the existing previous studies, particularly falling within the higher range of estimates reported for other Asian populations [22,23]. The prevalence of elevated depressive symptoms in our cohort (21.83%) was slightly higher than the previously reported average for China [40–42]. This elevated rate could be influenced by the unique socio-cultural context of Yunnan, as well as the overarching stress of the COVID-19 pandemic during the recruitment period [8,43]. However, as our internal comparison within the same pandemic context and the supportive animal data suggest, the depressive state itself appears to be the critical mediator of risk.

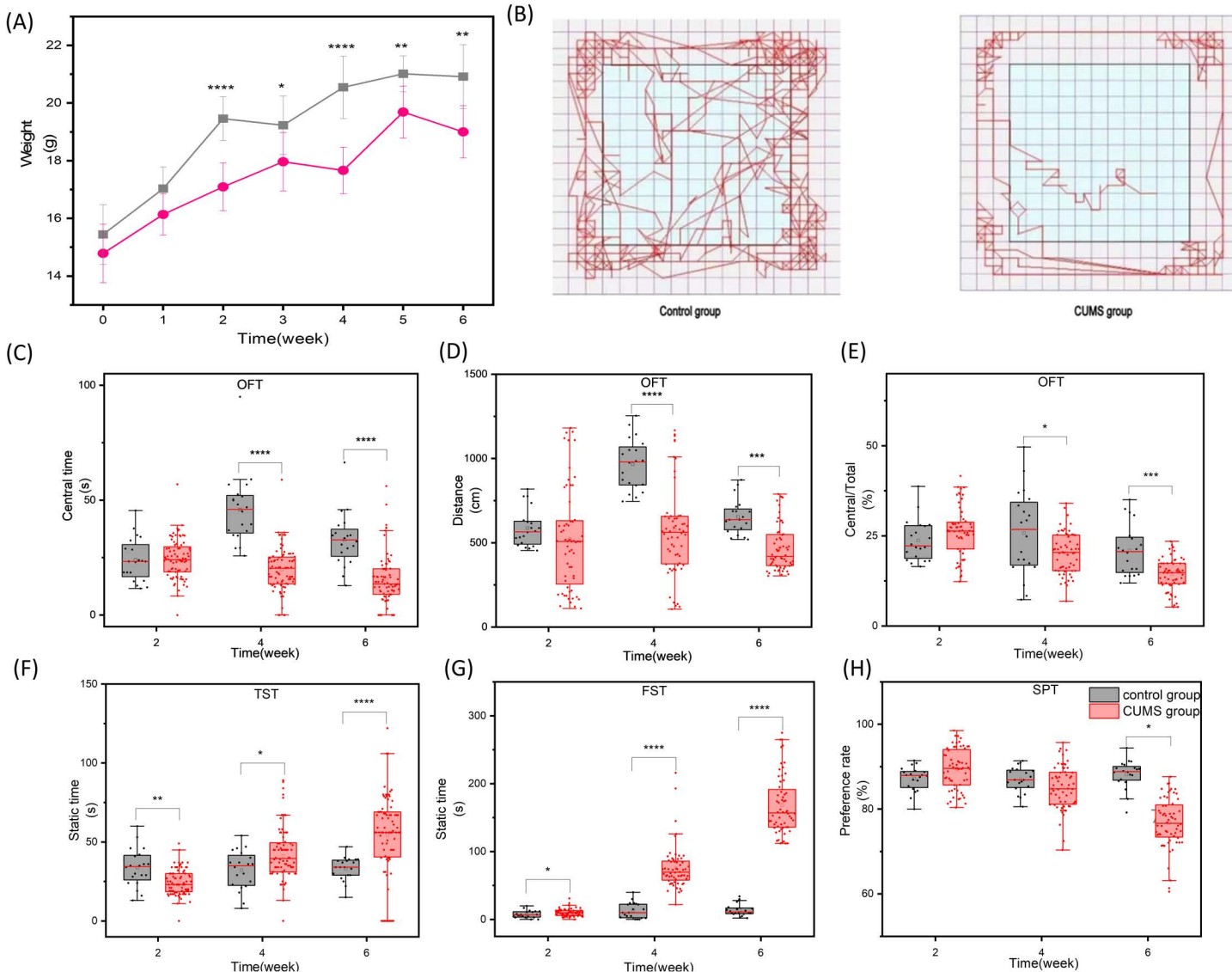

**Fig 4. Comparison of mouse behavior between CUMS group and control group.** (A) Mice in the CUMS group and control group were subjected to weight measurements once a week for 6 weeks. After two weeks of modeling, the body weight of the CUMS group was lower than that of the control group until week 6. (B) In the OFT, the activity trajectories of mice in the control group were more complex than those in the CUMS group, with longer movement distances and more time spent in the central region. (C) The central grid residence time in the CUMS group was significantly lower than that in the control group. (D) The total distance of exercise in the CUMS group was significantly lower than that in the control group. (E) The central grid/total distance traveled in the open field. * $P < 0.05$,** $P < 0.01$,**** $P < 0.0001$.

## Mechanistic interpretations

The biological plausibility of our findings is strong. The association may be mediated by a dysregulation of the hypothalamic-pituitary-adrenal (HPA) axis, leading to increased release of cortisol and inflammatory cytokines, which are known to promote uterine contractility and cervical remodeling [44–47]. Our CUMS mouse model, which isolates psychological stress from other human-specific confounders, provides direct experimental evidence for this pathway.

**Table 3. Pregnancy results of CUMS group and control group.**

| Variable | All (n = 48) | Control group (n = 16) | CUMS group (n = 32) | χ² | P value |
|---|---|---|---|---|---|
| Pregnancy outcomes | | | | 8.180 | 0.018 |
| PTB | 12 (25.00) | 0 (0.00) | 12 (37.50) | | |
| Non-PTB | 33 (68.75) | 15 (93.75) | 18 (56.25) | | |
| Pregnancy loss | 3 (6.25) | 1 (6.25) | 2 (6.25) | | |

## Potential mechanisms linking prenatal depression and preterm birth

The biological mechanisms underlying the association between prenatal depression and PTB are likely multifactorial, involving complex neuroendocrine and immune pathways. One prominent pathway is the dysregulation of the maternal HPA axis [46,47]. Chronic psychological stress and depression can lead to hypercortisolemia, and elevated cortisol may cross the placenta, potentially disrupting fetal development and triggering pro-inflammatory cascades that promote uterine contractility and cervical remodeling [47–49]. Concurrently, depression is associated with a heightened state of systemic inflammation, characterized by increased levels of pro-inflammatory cytokines such as IL-6 and TNF-α. These cytokines are known to play a direct role in the initiation of labor [47]. Our experimental model, which demonstrated a direct link between depression-like behavior and PTB in mice, provides a controlled platform supporting the plausibility of these biological pathways, independent of human sociocultural confounders. Future research integrating biomarker assessments in human cohorts is warranted to further elucidate these mechanisms.

## Context of the COVID-19 pandemic

Our study was conducted during the global COVID-19 pandemic, a period of unprecedented population-level stress. This context may have contributed to the overall prevalence of depressive symptoms we observed. However, the primary finding of an association between depressive symptoms and PTB is unlikely to be solely an artifact of the pandemic environment [50,51]. First, our study relied on an internal comparison; all participants were exposed to the same macro-stressor, yet the risk of PTB was significantly elevated only in those women who developed high levels of depressive symptoms. This suggests that individual psychological vulnerability, rather than the shared external environment per se, was the key factor associated with adverse outcomes. Second, and more definitively, our animal experimental findings provide direct supportive evidence. The induction of depression-like behaviors in mice under controlled, pandemic-free laboratory conditions led to a significantly higher incidence of PTB [52–54]. Together, these lines of evidence strengthen the plausibility of a direct contribution of prenatal depression to PTB pathophysiology.

## Strengths and limitations

The key strengths of our study include its prospective design, the combination of a human cohort with an experimental animal model, the use of a validated scale for depressive symptoms, and the adjustment for a comprehensive set of confounders across multiple statistical models.

This study also has limitations. First, depressive symptoms were assessed using the EPDS, a screening tool, and were measured only once during early pregnancy; repeated assessments across trimesters could provide a more dynamic understanding of their impact. Second, while we adjusted for numerous sociodemographic and clinical factors, residual confounding cannot be entirely ruled out. Notably, our characterization of the human study group was not exhaustive. Important factors known to influence both depression and pregnancy outcomes—such as detailed social support networks, history of trauma or adverse childhood experiences, partner relationship quality, and genetic predispositions—were

not measured and represent a potential source of unmeasured confounding. Third, related to the above, the moderate sample size of the subgroup with elevated depressive symptoms (n = 320) may limit the stability of estimates, particularly within finer strata. This is exemplified by the non-significant association observed for the severe depression subgroup (EPDS ≥19), where the very wide confidence interval (aRR = 1.67, 95% CI: 0.38–7.36) strongly suggests the result is likely due to a lack of statistical power rather than evidence of a true null effect. Consequently, the generalizability of our findings to other populations and to the most severe spectrum of depression requires verification in larger, more comprehensively characterized cohorts. Fourth, in the animal model, the use of tail handling for identification and during specific tests (instead of more refined methods like tunnel handling for all non-procedural activities) may have contributed to baseline stress levels. Future studies incorporating such advanced welfare techniques could further improve the reliability of behavioral outcomes.

Future research should, therefore, aim to integrate more comprehensive psychosocial and biological assessments within larger prospective cohorts and employ optimized animal handling protocols to better delineate the specific role of prenatal depression and the underlying mechanisms.

## Conclusion

In conclusion, our study provides robust evidence that prenatal depressive symptoms are a significant risk factor for PTB. These findings highlight the critical need to integrate mental health screening into routine prenatal care. Future research should focus on implementing and evaluating interventions aimed at mitigating depressive symptoms during pregnancy to improve maternal and child health.

## Supporting information

**S1 Fig. Comparison of birth numbers of mice in the CUMS group and the control group.**
(DOC)

**S1 Table. Univariate and Multivariate Analysis of Pregnant with Premature Birth.**
(DOC)

**S2 Table. Behavioral changes in the CUMS group (x ± s).**
(DOC)

**S3 Table. Pregnancy between CUMS and control group.**
(DOC)

**S4 Table. Weight in CUMS group and control group.**
(DOC)

**S1 File. OFT.Data.**
(XLSX)

**S2 File. SPT.Data.**
(XLSX)

**S3 File. TST.Data.**
(XLSX)

**S4 File. FST.Data.**
(XLSX)

## Acknowledgments

We thank Weizhou Wang for his assistance in writing the article.

## Author contributions

**Conceptualization:** Siguo Chen, Xinzi Xie.

**Data curation:** Siguo Chen, Qihan Wang.

**Formal analysis:** Siguo Chen, Guanghong Yan, Jinman Zhang.

**Funding acquisition:** Siguo Chen, Qihan Wang, Jinman Zhang, Hongying Li.

**Investigation:** Siguo Chen, Xinzi Xie, Hongying Li, Dingyun You.

**Methodology:** Xinzi Xie, Qihan Wang, Jie Zhong, Qian Wang, Hongying Li.

**Project administration:** Guanghong Yan, Hongying Li.

**Resources:** Guanghong Yan, Jie Zhong, Qian Wang.

**Software:** Jie Zhong.

**Supervision:** Xinzi Xie, Qian Wang, Jinman Zhang.

**Validation:** Jinman Zhang, Dingyun You.

**Writing – original draft:** Dingyun You.

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
