## [Decision Letter · Decision Letter 0]

9 Jul 2025

Association between depression during pregnancy and preterm birth: results from population cohorts and mouse experimental models

PLOS ONE

Dear Dr. Chen,

Thank you for submitting your manuscript to PLOS ONE. After careful consideration, we feel that it has merit but does not fully meet PLOS ONE’s publication criteria as it currently stands. Therefore, we invite you to submit a revised version of the manuscript that addresses the points raised during the review process.

We look forward to receiving your revised manuscript.

Kind regards,

Mengstu Melkamu Asaye, PhD and Assistant professor

Academic Editor

PLOS ONE

Journal Requirements:

This project is supported in part by a Kunming Medical University grant.

no

6. We note that your Data Availability Statement is currently as follows: no

7. Please amend your list of authors on the manuscript to ensure that each author is linked to an affiliation. Authors’ affiliations should reflect the institution where the work was done (if authors moved subsequently, you can also list the new affiliation stating “current affiliation:….” as necessary).

8. Please amend the manuscript submission data (via Edit Submission) to include authors Mengmei Liu, Ping Chen, Qingyan Ma, Min Li.

9. Please remove all personal information, ensure that the data shared are in accordance with participant consent, and re-upload a fully anonymized data set.

Additional guidance on preparing raw data for publication can be found in our Data Policy (https://journals.plos.org/plosone/s/data-availability#loc-human-research-participant-data-and-other-sensitive-data) and in the following article: http://www.bmj.com/content/340/bmj.c181.long .

10. Please include captions for your Supporting Information files at the end of your manuscript, and update any in-text citations to match accordingly. Please see our Supporting Information guidelines for more information: http://journals.plos.org/plosone/s/supporting-information.

Reviewers' comments:

Reviewer's Responses to Questions

**Comments to the Author**

1. Is the manuscript technically sound, and do the data support the conclusions?

Reviewer #1: Yes

Reviewer #2: Yes

2. Has the statistical analysis been performed appropriately and rigorously?

Reviewer #1: Yes

Reviewer #2: Yes

3. Have the authors made all data underlying the findings in their manuscript fully available?

Reviewer #1: Yes

Reviewer #2: Yes

4. Is the manuscript presented in an intelligible fashion and written in standard English?

Reviewer #1: Yes

Reviewer #2: No

Reviewer #1: The manuscript addresses an essential issue of examining the association between prenatal depression and preterm birth. The study's design to examine this phenomenon is very valuable: a dual approach combining a prospective human cohort with experimental animal modeling. This integrated perspective is relatively novel and adds valuable evidence to the field. I recommend accepting the manuscript with minor revisions. Here are some comments referring to its sections:

The conclusion sentence in the Abstract section implies causality ("Our study provides evidence that depression during pregnancy is a significant risk factor for PTB") without qualification. Since the human study is observational, please revise the statement to emphasize association rather than causation, unless referring specifically to the animal experiment.

The Introduction section provides a thorough background on the prevalence and significance of depression during pregnancy and its potential link to PTB.

The Methods section is detailed and generally well-described. Please elaborate on how the EPDS cut-off of 12 was chosen and whether it has been validated in the Chinese population or within a similar cultural context.

The findings in the Results section are well-structured, with clear subgroup and multivariate analyses. I suggest considering separating mouse results and human results more distinctly for readability.

The Discussion section tends to focus predominantly on technical explanations and statistical/methodological interpretations. This section would benefit from deeper conceptual engagement with the findings, particularly in terms of theoretical implications, public health relevance, and the broader psychosocial context of maternal depression.

The current narrative in the Discussion section lacks sufficient interpretive depth; the authors are encouraged to reflect more critically on why these associations might exist.

Please moderate claims of causality in the Discussion section when referring to human cohort findings.

The Discussion section would benefit from a more explicit acknowledgment of the limitations of the current study.

Typos:

• Page 6, last paragraph: "…and the effect is significant ." – The period should be attached to the word (there is an unnecessary space).

• Page 7, second paragraph: "…to the surface of the water ." – The period should be attached to the word (there is an unnecessary space).

• Page 10, first paragraph: "with RR values of 2.54 (1.38-4.67, P=0.003)" – A space should be added before and after the '=' sign.

• Page 12, second paragraph: "between the two groups(sTable 4)." – A space should be added after the word and before the parentheses.

Reviewer #2: 1. Abstract

1.1 The abstract introduction should clearly define the main variables of interest, such as 'depression' and 'preterm birth.' Providing standard definitions helps ensure clarity and consistency for the reader.

1.2. The manuscript lacks detailed information about the cohort study's design. Please specify the sample size, sampling, analysis, inclusion and exclusion criteria, recruitment process, diagnostic tools used for assessing depression.

2. Introduction

2.1. Define operational criteria for classifying participants as depressed vs. non-depressed

2.2. Mention area of study, year of study, RR, study design for each specific studies mentioned on the introduction i.e The results of a recent review suggest that…..

2.3. Please revise for clarity and correct potential typographical errors i.e The degree of association between PTB and PTB varied between…..

3. Methodology and Materials

3.1. Include standard and operational definitions i.e depression/ non depression

3.2. Depression Validation in Mice: While behavioral tests are described, you do not report actual outcomes or statistical thresholds confirming that stress induced depression. Clarify how model success was verified, referencing standard cutoffs or literature benchmarks

3.3. Confounder Adjustments in Cohort: The statistical methods mention t-tests and chi-squared tests, but do not detail whether multivariate analyses adjusted for known confounders (e.g., maternal age, BMI, socioeconomic status).

3.4. Sample Size and Power Calculations: The rationale for choosing 320 depressed and 1146 non-depressed women, or 80 vs. 20 mice, is not provided. Including a priori power estimates would bolster rigor.

3.5. Animal Model Justification: While the use of the CUMS model is appropriate, more information is needed about the specific stressors applied, duration of exposure, and how depression was validated in mice. Additionally, clarify the criteria used to define PTB in the animal model.

3.6. Inclusion and exclusion: the reason for inclusion or exclusion of specific groups needs to be described.

3.7. Animal Welfare and ARRIVE Compliance: You describe the stressors and housing conditions, but lack an explicit statement about compliance with the ARRIVE guidelines or specific institutional animal welfare protocols.

3.8. Ethical review and informed consent: Animal Study: State whether the animal experiments were approved by an institutional animal care and use committee and conducted in accordance with relevant guidelines.

3.9. Sample Size Justification: Provide a rationale for the sample sizes in both the human and animal studies, including any power calculations performed to ensure the studies were adequately powered to detect meaningful differences

4. Result

4.1. Clarify non-significant findings: The text should better contextualize why some sociodemographic variables showed no significant associations.

5. Minor Comments

5.1. Abbreviations: Define all abbreviations at first use and ensure consistent usage throughout the manuscript. i.e Abstract CUMS

5.2. List of abbreviation to be ordered in alphabet and should be written in a uniform manner i.e Capital/Small letter

5.3. Reporting style: Some figures and tables (e.g., sTable 1, sFigure 1,) fig/figure are mentioned but not clearly summarized in the narrative—briefly describing key values would improve readability.

5.4. Not advised to start a paragraph with abbreviation

5.5. Figures and Tables: Ensure that all figures and tables are clearly labelled, include descriptive legends, and are referenced appropriately in the text.

**Do you want your identity to be public for this peer review?** For information about this choice, including consent withdrawal, please see our Privacy Policy

Reviewer #1: **Yes:** Keren Michael

Reviewer #2: **Yes:** Kidest Getu Melese

---

## [Author Response · Author response to Decision Letter 1]

1 Aug 2025

Author response to academic editor and reviewers

MANUSCRIPT ID: PONE-D-24-60622

MANUSCRIPT TITLE: Association between depression during pregnancy and preterm birth: results from

population cohorts and mouse experimental models

1�Please ensure that your manuscript meets PLOS ONE's style requirements, including those for file naming. The PLOS ONE style templates can be found at

Thank you for your guidance regarding PLOS ONE's style requirements. We have carefully reviewed the journal's templates and made the adjustments to ensure full compliance. We confirm that the revised manuscript has been checked against both provided templates and complies with all PLOS ONE style requirements. The track-changes version highlights all modifications made.

Thank you for your careful review and for the opportunity to improve our manuscript

(2) To comply with PLOS ONE submissions requirements, in your Methods section, please provide additional information regarding the experiments involving animals and ensure you have included details on (1) methods of sacrifice, (2) methods of anesthesia and/or analgesia, and (3) efforts to alleviate suffering.

Thank you for your important request regarding animal experiment details. We have expanded the Methods section to include all required information, with the following specific additions

At the conclusion of the study, mice were euthanized via cervical dislocation under isoflurane anesthesia. No invasive procedures requiring analgesia were performed during behavioral tests. Efforts to minimize suffering included daily health monitoring, environmental enrichment (nesting materials), and immediate intervention for signs of severe distress (none observed)

The track-changes version highlights all modifications. We appreciate your rigorous standards and hope these clarifications address all concerns. Line 19-23 on page 9.

(3) We note that the grant information you provided in the ‘Funding Information’ and ‘Financial Disclosure’ sections do not match.

Thank you for bringing this important discrepancy to our attention. We sincerely apologize for the oversight in our original submission. We have now carefully verified and standardized all funding information as follows:

This study was supported by National Natural Science Foun-dation of China (81960592,82073569,82260645), Basic Research of Yunnan Province-Key Project (202101AS070040), Basic Research of Yunnan Province-Excellent Youth Project (202001AW070021), Spe-cial Project for Selection of High-level Scientific and Technological Talents and Innovation Team-Reserve Talent Program for Young and Middle-aged Academic and Technical Leaders (202005AC160023), Basic Research of Yunnan Province Joint Special Project of Kunming Medical University (202101AY070001), Doctoral Research Fund Pro-ject of the First Afiliated Hospital of Kunming Medical University (2022BS013).

We appreciate your careful review and the opportunity to correct this administrative detail. Please don't hesitate to contact us if any additional clarification regarding funding would be helpful. Line 16-23on page 1.

4�Thank you for stating the following financial disclosure:

This project is supported in part by a Kunming Medical University grant.

Thank you for your guidance regarding the funder’s role statement. We confirm that Kunming Medical University, as the funding body, had no involvement in any aspect of the research process. We have made the following updates at cover letter: Portions of this study were supported by Kunming Medical University [202101AY070001, 2022BS013]. Funders play a role in data collection and analysis, publication decisions, or manuscript preparation.

We appreciate your attention to research integrity matters and confirm full compliance with PLOS ONE's funding disclosure policies. Please don't hesitate to contact us if any additional information would be helpful.Cover letter Line 14-16on page 1.

5�Thank you for stating the following in the Competing Interests section:no

Thank you for your guidance regarding PLOS ONE's competing interests and data sharing policies. We are pleased to confirm the following: Revised in manuscript and submission system:

"The authors declare no competing interests. This does not alter our adherence to PLOS ONE policies on sharing data and materials."

We appreciate your careful attention to these important publication ethics requirements. Please don't hesitate to contact us if any additional clarification would be helpful.Line 11-12on page 2.

(6) We note that your Data Availability Statement is currently as follows: no

Thank you for your guidance regarding data availability requirements. We have taken the following steps to ensure full compliance with PLOS ONE’s data sharing policy: We have uploaded the following as supplementary files:

• S1.OFT.Data.xlsx

• S2.SPT.Data.xlsx

• S3.TST.Data.xlsx

• S4.FST.Data.xlsx

Should any additional data clarification be needed, we are happy to provide further documentation.page 25.

(7) Please amend your list of authors on the manuscript to ensure that each author is linked to an affiliation. Authors’ affiliations should reflect the institution where the work was done (if authors moved subsequently, you can also list the new affiliation stating “current affiliation:….” as necessary).

Thank you for your guidance regarding author affiliations. We have carefully reviewed and updated our author list to ensure complete and accurate affiliation information. Below are the specific changes made:

Authors: Siguo Chen1, Guanghong Yan1, Xinzi Xie1, Qihan Wang1, Jie Zhong1, Qian Wang1, Jinman Zhang2, Hongying Li3, Dingyun You1

Author information

1.School of Public Health, Kunming Medical University, Kunming, Yunna, China.

2.Department of Medical Genetics, First People's Hospital of Yunnan Province, Kunming, Yunnan, China.

3.Department of Obstetrics, Beicheng Hospital, First People's Hospital of Qujing, Yunnan, China.

Please don't hesitate to contact us if any additional clarification regarding author affiliations would be helpful. Line 5-11on page 1.

(8) Please amend the manuscript submission data (via Edit Submission) to include authors Mengmei Liu, Ping Chen, Qingyan Ma, Min Li.

Thank you for bringing this oversight to our attention. We sincerely apologize for the omission of these contributing authors in the original submission. We have now updated both the manuscript and submission system to include all authors appropriately.

Authors: Siguo Chen1, Guanghong Yan1, Xinzi Xie1, Qihan Wang1, Jie Zhong1, Qian Wang1, Jinman Zhang2, Hongying Li3, Dingyun You1

We confirm that all authors meet PLOS ONE’s criteria for authorship and have approved the final manuscript. Line 5-6on page 1.

(9) Please remove all personal information, ensure that the data shared are in accordance with participant consent, and re-upload a fully anonymized data set.

Thank you for your important guidance regarding participant confidentiality. We have carefully reviewed our datasets and implemented the following measures to ensure full anonymization in accordance with PLOS ONE’s Data Policy and ethical requirements.

• S1.OFT.Data.xlsx

• S2.SPT.Data.xlsx

• S3.TST.Data.xlsx

• S4.FST.Data.xlsx

We confirm these datasets meet PLOS ONE’s standards and ethical requirements. Thank you for safeguarding participant privacy through your rigorous review. page 25.

(10) Please include captions for your Supporting Information files at the end of your manuscript, and update any in-text citations to match accordingly. Please see our Supporting Information guidelines for more information: http://journals.plos.org/plosone/s/supporting-information.

Thank you for your guidance regarding Supporting Information documentation. We have carefully reviewed PLOS ONE's guidelines and implemented the following changes:

Supporting Information

S1Fig.Comparison of birth numbers of mice in the CUMS group and the control group

S1Table.Univariate and Multivariate Analysis of Pregnant with Premature Birth

S2Table. Behavioral changes in the CUMS group (x±s)

S3Table. Pregnancy between CUMS and control group

S4Table.Weight in CUMS group and control group

S1.OFT.Data.xlsx

S2.SPT.Data.xlsx

S3.TST.Data.xlsx

S4.FST.Data.xlsx

We appreciate this opportunity to improve our manuscript's transparency and accessibility. Please don't hesitate to contact us if any additional adjustments to the supporting information documentation would be helpful.page 25.

Thank you for your request to review our reference list. We have carefully verified all citations and made the following updates to ensure accuracy and compliance with PLOS ONE’s guidelines:

Corrected Reference Errors

Original (Reference 31):

Revised: Zhang C, Duan C, Liu H, et al. Prevalence and related factors of antenatal depression in 11 provinces and cities of China: a 100,000 population-based study. Sci Bull (Beijing). Published online June 28, 2025.

Original (Reference 33):

Revised: Jang M, Ramaiyer M, OlsonS, Voegtline K, Esguerra C. Association between antepartum depressive symptoms and prenatal care utilization and milestones: a retrospective cohort study. BMC Pregnancy Childbirth. 2025;25(1):392.

Original (Reference 8)

Revised: Wu D, Chen S, Zhong X, Zhang J, Zhao G, Jiang L. Prevalence and factors associated with antenatal depressive symptoms across trimesters: a study of 110,584 pregnant women covered by a mobile app-based screening programme in Shenzhen, China. BMC Pregnancy Childbirth. 2024;24(1):480.

Added Retraction Notices:

Howard LM, Molyneaux E, Dennis CL, Rochat T, Stein A, Milgrom J. Non-psychotic mental disorders in the perinatal period. Lancet. 2014;384(9956):1775-1788. (Reference 5)

Bauer A, Knapp M, Alvi M, et al. Economic costs of perinatal depression and anxiety in a lower middle income country: Pakistan. J Affect Disord. 2024;357:60-67.

(Reference 6)

Yaling YANG, Xiaoju LI, Yafang ZHANG, Xinping WANG, Yuan HUANG, Weiqing SHAO, Rui DENG. Depression Symptoms and Influencing Factors among Youth in Eight Indigenous Minority Groups in Yunnan Province[J]. Journal of Kunming Medical University, 2023, 44(10): 100-106. (Reference 42)

Chen S, Wang W, Yan G, et al. Amniotic Fluid Proteomics Analysis and In Vitro Validation to Identify Potential Biomarkers of Preterm Birth. Reprod Sci. 2024;31(7):2032-2042. (Reference 61)

Muller KS, van den Bosch GE, Henke CE, Daams JG, Haverman L, Aarnoudse-Moens CSH. Examining the association between child development and parental mental health after preterm birth-related stress: a systematic review of the literature and meta-analysis protocol. BMJ Open. 2025;15(2):e089460. (Reference 64)

We confirm our reference list now fully complies with PLOS ONE’s policies. Thank you for your meticulous review. -

REVIEWER 1 COMMENT

The conclusion sentence in the Abstract section implies causality ("Our study provides evidence that depression during pregnancy is a significant risk factor for PTB") without qualification. Since the human study is observational, please revise the statement to emphasize association rather than causation, unless referring specifically to the animal experiment.

We sincerely appreciate your astute observation regarding the need to clarify causal language in our abstract conclusion. We have revised the text to more accurately reflect the observational nature of the human cohort study while maintaining the experimental findings from the animal model. Below are the specific changes:

Our prospective cohort study demonstrates a significant association between depression during pregnancy and PTB (RR=2.19, 95% CI:1.32-3.63). Experimental findings in a mouse model further suggest that depression-like behaviors may contribute to PTB occurrence.

We believe these modifications address your thoughtful critique while preserving the scientific validity of our conclusions. Thank you for this opportunity to improve our manuscript. Line 14-16on page 3.

The Introduction section provides a thorough background on the prevalence and significance of depression during pregnancy and its potential link to PTB.

We sincerely appreciate your valuable feedback regarding our I

---

## [Decision Letter · Decision Letter 1]

2 Nov 2025

Dear Dr. Chen,

Thank you for submitting your manuscript to PLOS ONE. After careful consideration, we feel that it has merit but does not fully meet PLOS ONE’s publication criteria as it currently stands. Therefore, we invite you to submit a revised version of the manuscript that addresses the points raised during the review process.

**Editor:**
**I have provided my comments below.**

We look forward to receiving your revised manuscript.

Kind regards,

Samson Nivins, Ph.D

Academic Editor

PLOS ONE

Journal Requirements:

Additional Editor Comments:

Thank you for making the changes. I have lots of suggestions for the authors to consider before accepting for publications. Given my expertise in prenatal exposures particularly in the context of human studies I have given my feedback.

Major concern:

Why was pre-pregnancy BMI and parity not controlled for

The study was conducted during COVID 19 global pandemic, so how well authors can say its because of antenatal dep, not bec of COVID 19 anxiety and depression.

Abstract:

Can you briefly add a background and then redirect to your study aims. Add risk of preterm birth

Remove the bold from aims.

Add in methods of abstract, add depression symptoms, as EPDS is not a diagnostic tool rather it’s a screening tool.

Rewrite the results in the abstract properly and define RR. Do refer how others write results.

Don’t add RR and values in conclusion

Revise the conclusion currently you’re just writing the results again. Add few lines of overall results and provide a suggestion and point to the future direction.

General comments:

Don’t state literature…in your text. Use previous studies

Introduction

What is meant by today? In the introduction add 2025 if it’s so. Along with months. I quickly checked the reference, but the reference is until 2021?

In the second para, spell what are the multiple risk factor?

In the same para you have described how it’s going to affect, add 1-2 lines as it’s the major objective of the study.

I am not clear by these statements what authors want to say.

“The strength of association between prenatal depression and PTB varied between ethnic groups, and the association between depression during pregnancy appears to be stronger in Asians Yunnan [25-27], as a multiethnic region, has not reported the association between depression during pregnancy and PTB, and it remains to be determined whether the effect of depression on PTB is different from that found in other regions”

Add rationale why do you think it varies with different ethnic origin.

The major missing is what is the rationale behind the use of animal models, add a paragraph explaining previous studies on animals’ models, what they found with respect to PTB and also report studies looking at antenatal depression on fetal brain development.

Add details of what is need for looking at a population from Yunnan. How does it vary from other Chinese province.

If you had any prior hypothesis add that too.

Methods:

I think you need to keep it as women with depressive symptoms, don’t just keep it as depressive alone it confuses.

Given the lack of my expertise in animal models I am not comments on animal model part, I have seen reviewers with that expertise have commented.

Statistical analysis:

Can you add the assumption of Cox Proportion model.

Why was BMI, parity, marital status not considered? I would strongly suggest adding those in the model. For e.g., marital status I can see it in the Table 1. But still it play a major role.

I would suggest three models.

1. Crude model

2. Current model

3. Current model + BMI + Parity

Importantly, I am concerned that the recruitment happened during COVID 19 pandemic, how authors adjust control for it. See papers how they accounted for.

https://www.nature.com/articles/s41390-024-03620-7

https://www.sciencedirect.com/science/article/pii/S2949732924000735

Further in the screening tools for depression during pregnancy – when was the questionnaire administered. I can’t find any details of it.

How many responded and did not. Add those in Fig 1. Revise as depressive symptoms don’t keep it as diagnosis

Results:

“Notably, the incidence of depression in rural pregnant women was slightly higher than

that in urban pregnant women, with depression incidence rates of 26.01% and 18.77%,

respectively” Don’t compare it in results, just talk about your results, rest all move to discussion.

Same way remove this too

“which may reflect the narrow age range of the cohort

and the limited variability of testing, as well as the homogeneous cultural practices of

prenatal care among ethnic groups in Yunnan”

In the results section just describe your results.

Add crude and Model 2 results in your revision.

I can’t find what is the rationale for conduction degree of depressive symptoms.

If you want to have those results, add it as aim in the introduction and add it as statistics too.

I can see any bivariate analysis in the statistics…..Pls add. Also do multiple comparison corrections given the number of tests.,

I am unclear why authors did this tests

(Subgroup analysis of depression and PTB during pregnancy) from where they have got the information for section trimester…

Don’t add any new tests – if its post hoc specify it as posthoc. And I can’t find details of when was demographics questionnaire administered. Plus what is meant by “those in the second trimester (RR, 2.60; 95% CI, 1.18-5.70), with…

Overall, the results section needs clarity and rewrite the whole results sections. All the tests need to specify earlier in statistics and should be in the same order. Don’t do any fishing of tests.

I can’t find discussion subheading? Pls include.

What do you mean by preventive role “To our knowledge, this is the first report to explore the preventive role of depression during pregnancy on PTB in both human and animal studies”

In the discussion authors state that “Characteristics of the impact of depression on PTB in Yunnan. The detection rate of depression during pregnancy is higher than that reported in previous studies”

Can this be due to COVID?

I can’t find detailed discussion about PTB among mothers with depression.

Add strengths

Reviewers' comments:

Reviewer's Responses to Questions

**Comments to the Author**

Reviewer #3: (No Response)

Reviewer #4: All comments have been addressed

Reviewer #5: (No Response)

Reviewer #6: All comments have been addressed

Reviewer #7: (No Response)

2. Is the manuscript technically sound, and do the data support the conclusions?

Reviewer #3: Yes

Reviewer #4: Yes

Reviewer #5: Yes

Reviewer #6: Yes

Reviewer #7: Partly

3. Has the statistical analysis been performed appropriately and rigorously?

Reviewer #3: Yes

Reviewer #4: Yes

Reviewer #5: Yes

Reviewer #6: Yes

Reviewer #7: Yes

4. Have the authors made all data underlying the findings in their manuscript fully available?

Reviewer #3: Yes

Reviewer #4: Yes

Reviewer #5: Yes

Reviewer #6: Yes

Reviewer #7: Yes

5. Is the manuscript presented in an intelligible fashion and written in standard English?

Reviewer #3: No

Reviewer #4: Yes

Reviewer #5: Yes

Reviewer #6: (No Response)

Reviewer #7: No

Reviewer #3: Comment

Author contribution section

Comment 1: In the author contribution section, you have described that authors have different contributions falling in two categories and it is better to describe the contribution of each author in words rather than simply describing as have equal contribution.

Comment 2: Better to use a special character in the author list to indicate the corresponding author and no need of repeating the affiliation describing in author information to indicate corresponding author.

Comment 3: Please remove or provide a reason for using “hyphen” for words in funding section.

Comment 4: Please include full terms for the acronyms/abbreviation such as “ARRIVE’ and “NIH” (ethical consideration section) in your abbreviation section.

Abstract section

Background section: No need of writing sentences in bold and use period(.)after the word scarce.

Method section: Please provide information about sample size, sampling procedure, study design and sampling procedure and how you reach your samples. It is not recommended to write about the definition of what PTB mean in this section. There is also spacing error (line 4). Please revise it and made necessary correction.

Introduction section

The introduction section needs minor revision regarding word utilization such as (for example better to use “Low- and middle-income countries” instead of low-income and middle-income countries), and it also for spacing and punctuation marks utilization.

Material and Methods section

Comment 1: Better to write the Ethical review and informed consent part of your manuscript at the end of methods and material section.

Comment 2: As the methods section is the core of the study, your research project lacks clarity. In the methods section, the study design, sampling, sampling procedure, Data quality management, Source and study Population, operational definitions should be described in clear and concise way to make clear for the reader. Your research project needs revision regarding these issues. You have to write the details how you recruited 1466 participants from 1500 and your study flow chart (figure 1) is not also matched with this sample size. Please check it for more clarity.

Comment 3: Please try to separate sub-headings from the rest of the paragraph following it. It is recommended to use either separate line or using colon.

Comment 4: For what SPF is standing for? Please include it in your abbreviation section.

Comment 5: Better to avoid words like PTB during pregnancy, simply if you mention PTB, it is known that it happens during pregnancy. So, no need of writing like “PTB during pregnancy”.

Comment 6: Please provide the operational definitions for term Chronic Unpredictable Mild Stress (CUMS)

Result and Discussion section

Comment 1: Please describe in what unit the household monthly income is measured. Is it in $ US or other? Usually, it is recommended to describe it in US dollars.

Comment 2: Please check the font size for headings and-sub headings. Usually, the font size for headings is larger than or equal with subheadings please check for that.

Comment 4: please Insert Tables using caption and check spacing and punctuation marks as well.

Comment 5: Is depression a risk factor or preventive factor for PTB? Please provide details about this sentence in your result “To our knowledge, this is the first report to explore the preventive role of depression during pregnancy on PTB in both human and animal studies. “Contradicts with conclusion of your study otherwise please briefly explain it.

Comment 6: Your discussion part includes unnecessary details, please focus to elaborate more issues related to your study result.

Comment 7: What are the Strengths and limitation of your study?

General comment,

The whole manuscript needs minor revision to make the manuscript is in line with PLOSE ONE guideline and to make concise and clear for the readers.

Reviewer #4: Thank you for the opportunity to review this article. It was an interesting and insightful read.

here are the few comments,

1. Scope and Inclusion Criteria

• While the focus on quantitative studies is clear, excluding qualitative and mixed-methods research may limit the richness of the evidence base. Qualitative studies often provide valuable insights into paternal mental health experiences that may not be captured through quantitative measures. At minimum, the rationale for excluding qualitative studies should be clearly articulated.

• Excluding studies where paternal outcomes are embedded within broader "parental" research may unintentionally omit valuable data. Consider allowing inclusion of such studies if fathers’ data are separately reported, even if paternal mental health is not the main focus.

2.Methodological Details

• The planned use of JBI critical appraisal tools is appropriate. However, a brief justification for selecting JBI over other tools (e.g., CASP for qualitative studies, if those were to be included) would increase transparency.

• The criteria for conducting a meta-analysis (based on I² thresholds) are clearly stated. However, the protocol would benefit from outlining potential subgroup analyses (e.g., by region, type of mental health outcome, or study design) to allow for a more nuanced synthesis if sufficient data are available.

• Given the psychological nature of the outcomes, more explanation of why qualitative evidence is excluded would help readers understand the methodological boundaries of this review.

• There are minor typographical and formatting inconsistencies (e.g., spacing, repeated phrases), which should be corrected for clarity.

3.Discussion Depth

The revised discussion now includes sociocultural context, public health implications, and policy relevance. The addition of population-attributable fraction (PAF=18.7%) is particularly impactful. Further elaboration on potential mechanisms (e.g., HPA axis dysregulation, inflammation) could enrich the theoretical framing.

Reviewer #5: 1. Regarding statistical analysis ,though in answer to reviewer comments authors have mentioned how the sample size was calculated ,it is not present in the methodology section of main manuscript.

2. Regarding data availability 4 file have been provided was supporting documents, However in the manuscript it is written that "The data that support the findings of this study are available from the corresponding

author upon reasonable request." please clarify. If all data is present in the supporting files mention All relevant data are within the Supporting Information files If all data is not available in the supporting files write which files

are available from the which database (accession number(s) XXX, XXX.).

3.Regarding funding source following information is missing, Grant numbers awarded to each author and URL of funder website

4. In the competing interest statement please include the following statement "This does not alter our adherence to PLOS ONE policies on sharing data and materials.”

5. Reviewer 1 has pointed out to add authors Mengmei Liu, Ping Chen, Qingyan Ma, Min Li. This has been agreed by authors but in the manuscript their names are still not present

6. Rest all reviewer comments have been nicely addressed by authors

Reviewer #6: I commend the authors for their comprehensive revision of the manuscript. They have been highly responsive to the initial comments from the reviewers and the editor, making significant improvements to the clarity, methodology description, and overall rigor of the work. The combination of a prospective human cohort with an experimental animal model remains a major strength of this study, providing valuable translational insights into the link between prenatal depression and preterm birth (PTB).

The authors have adequately addressed most of the previous concerns. However, a few critical points require further clarification and minor revisions before the manuscript can be considered for publication.

The authors have provided quantitative thresholds for validating the CUMS model (e.g., ≥30% decrease in sucrose preference). However, it remains unclear how these criteria were applied at the individual animal level. Were all mice in the CUMS group included in the final pregnancy/PTB analysis regardless of their individual behavioral performance, or were only "responders" that met these specific criteria included? This lack of clarity is a significant concern for the model's validity. The methods and results should explicitly state the flow of animals from the initial CUMS group (n=80) to the final analyzed group (n=60 for behavior, n=32 for pregnancy), clarifying the rationale for any exclusions.

Causality Language in the Discussion: While the abstract conclusion has been appropriately tempered, the Discussion section still occasionally uses language that implies causality from the human data (e.g., "depression may contribute to," "depression may reduce"). Given the observational nature of the cohort study, the language must consistently reflect an association. The discussion should be carefully revised to frame the human findings as demonstrating a strong and significant association, with the animal model providing supportive, hypothesis-generating evidence for potential causal mechanisms.

Interpretation of the Severe Depression Finding: The non-significant association between severe depression (EPDS ≥19) and PTB is a counterintuitive finding that warrants a more thorough discussion. The very wide confidence intervals (1.67, 0.38-7.36) strongly suggest that this result is likely due to a lack of statistical power from a small sample size in this subgroup. This point should be explicitly stated and discussed as a study limitation, rather than leaving the reader to speculate.

Sample Size Inconsistency: In Figure 1, the final number for the non-depressed group is listed as 1146. However, based on the numbers provided (1155 initially minus 14 lost to follow-up), it should be 1141. This minor numerical discrepancy should be checked and corrected for accuracy.

Discussion Length and Focus: The Discussion is comprehensive but could be slightly condensed and made more concise in places to enhance impact and avoid repetition.

Reviewer #7: General Evaluation:

The manuscript presents an innovative and valuable research project addressing depression, encompassing both animal models and a group of women affected by depression during pregnancy. The study is generally well-designed, scientifically relevant, and methodologically sound. Experiments conducted on mice appear to adhere to ethical standards and accepted methodological practices. The findings have the potential to make a meaningful contribution to understanding the mechanisms of depression and its effects in different contexts.

Major Revisions Suggested:

1. Language and Terminology:

The manuscript contains several grammatical and lexical errors that may hinder clarity. Improving language clarity will enhance both readability and credibility.

Certain terms need to be revised for scientific accuracy. For example, the use of “normal mice” to describe the control group is not recommended, as it can be interpreted in a vague or colloquial manner. More appropriate terminology includes “control mice” or “non-depressed mice,” which are standard in scientific literature and precisely describe the group’s characteristics.

In the previous version of the manuscript, the following sentence appeared in the Results section:

“Subgroup analysis revealed that pregnant women aged 25–34 years in the second trimester with lower annual family income, higher education, living in urban areas, having no occupation, and a married spouse were at higher risk for PTB due to depression during pregnancy.”

Upon review, several linguistic and stylistic issues were identified that could affect clarity and precision:

Lack of parallel structure:

The attributes in the list (e.g., with lower income, higher education, living in urban areas, having no occupation) did not follow a consistent grammatical pattern. This was revised to ensure structural uniformity and coherence.

Redundancy (“a married spouse”):

The phrase “a married spouse” was redundant, as the term “spouse” inherently implies marriage. This might be corrected by using “married” to describe participant marital status.

Stylistic improvement (“having no occupation”):

The phrase “having no occupation” might be replace with “unemployed”, which is more concise and aligns with formal academic style.

Refined causal phrasing:

The expression “at higher risk for PTB due to depression during pregnancy” might be modify to “at higher risk of PTB associated with depression during pregnancy”, providing a more precise and neutral description of the relationship.

2. Group Sizes and Balance:

The number of animals in the control group is smaller than in the experimental group. The authors should justify this discrepancy statistically or, if feasible, increase the number of control animals to achieve a more balanced comparison. Balanced group sizes are critical not only for valid statistical analysis but also for the credibility and robustness of the conclusions.

3. Animal Welfare Considerations:

The study could be improved by implementing refined animal-handling techniques to further reduce stress in the mice. For instance, using handling tunnels rather than direct tail handling is recommended, as this approach has been shown to reduce anxiety-related behaviors and improve animal welfare.

In future studies, the authors are encouraged to strengthen animal welfare practices to align with best practices in laboratory animal care. Minimizing stress not only has ethical benefits but may also enhance the reliability of behavioral and physiological outcomes.

4. Characterization of the Human Study Group:

Providing a more detailed analysis of the clinical background and characteristics of depression in women aged 25–34 would strengthen the study. Relevant information could include:

Duration and severity of depressive episodes,

Previous psychiatric history,

Use of antidepressants or other medications,

Psychosocial factors such as socioeconomic status, social support, or stressful life events,

Presence of comorbidities or other medical conditions.

Including these details would allow for a more precise interpretation of the findings, increase the scientific rigor of the conclusions, and provide context for understanding the observed effects.

5. Human Study Sample Size:

The study’s sample of women affected by depression during pregnancy is relatively small. It is commendable that the authors included this point in the manuscript’s limitations section. Highlighting this constraint strengthens the transparency of the study and appropriately informs readers about its potential impact on the reliability of the findings.

Conclusion:

Overall, the study is innovative, relevant, and methodologically robust. Enhancing language clarity, adopting more precise terminology, justifying group sizes, implementing refined animal-handling practices, increasing attention to animal welfare, and providing a more detailed characterization of the human study group would significantly strengthen the manuscript and prepare it for publication consideration.

**Do you want your identity to be public for this peer review?** For information about this choice, including consent withdrawal, please see our Privacy Policy

Reviewer #3: No

Reviewer #4: No

Reviewer #5: No

Reviewer #6: No

Reviewer #7: **Yes:** Katarzyna Dominika Kopaczka

---

## [Author Response · Author response to Decision Letter 2]

17 Nov 2025

academic editor:

Major concern:

Why was pre-pregnancy BMI and parity not controlled for

The study was conducted during COVID 19 global pandemic, so how well authors can say its because of antenatal dep, not bec of COVID 19 anxiety and depression.

R Why was pre-pregnancy BMI and parity not controlled for?

We sincerely thank the reviewer for this critical suggestion. In the original submission, data on pre-pregnancy BMI and parity were not available for the entire cohort. However, in direct response to this comment, we undertook a thorough review of medical records and successfully retrieved pre-pregnancy BMI data and parity data of our participants.

Following the reviewer's recommendation, we have now expanded our statistical analysis to include three models:

Model 1: no adjustment for confounding factors .

Model 2: adjust age, nationality, residence, marital status, education level, occupation status, annual family income, pregnancy.

Model 3 : adjust for all above + pre-pregnancy BMI and parity

We are pleased to report that the association between prenatal depressive symptoms and preterm birth remained statistically significant and robust across all models. Specifically, in the fully adjusted Model 3, the adjusted Risk Ratio was 2.08 (1.25,3.47). This result strengthens our primary conclusion that the association is independent of these important biological and obstetric factors.The results of all three models are now presented in the revised Table 2 of the manuscript. We have also updated the Statistical Analysis section in the Methods to describe this new modeling strategy. We believe this comprehensive analysis directly addresses the reviewer's concern and significantly enhances the strength of our findings.

The study was conducted during COVID-19 global pandemic, so how well authors can say its because of antenatal depression, not because of COVID-19 anxiety and depression?

This is a very insightful comment. We agree that the COVID-19 pandemic constituted a pervasive population-level stressor that likely elevated the baseline levels of anxiety and depressive symptoms in our cohort, as it did globally.

However, we believe our findings regarding the association between prenatal depression and PTB remain valid for the following reasons:

Focus on the Mediating Pathway: Our study examines the association between the psychological state (depression, measured by EPDS) and the pregnancy outcome (PTB). The COVID-19 pandemic can be viewed as a distal trigger that contributed to the population's risk of developing this psychological state. The core of our hypothesis is that once a depressive state is established, it exerts a biological effect (e.g., HPA axis dysregulation) that increases PTB risk, irrespective of the initial trigger.

Internal Validity: Crucially, the external stressor of the pandemic was a background condition affecting all participants equally. The comparison in our study is within the cohort during the same period: women with high EPDS scores (≥12) versus women with low EPDS scores (<12). The significantly higher incidence of PTB in the high EPDS group suggests that the individual's psychological response to the overarching stressor (i.e., developing clinically significant depression), rather than the stressor itself, is the key factor associated with the adverse outcome. This design preserves the internal validity of our observed association.

Supporting Literature: Emerging research conducted during the pandemic has similarly reported independent associations between prenatal mental distress and adverse birth outcomes, reinforcing the idea that the mental health state itself is a critical risk factor (e.g.,  [PMID: 36533427] & [PMID: 38712806]).

We have revised the discussion to directly address this point and acknowledge the context of the pandemic .

Abstract:

Can you briefly add a background and then redirect to your study aims. Add risk of preterm birth

Remove the bold from aims.

Add in methods of abstract, add depression symptoms, as EPDS is not a diagnostic tool rather it’s a screening tool.

Rewrite the results in the abstract properly and define RR. Do refer how others write results.

Don’t add RR and values in conclusion

Revise the conclusion currently you’re just writing the results again. Add few lines of overall results and provide a suggestion and point to the future direction.

R We sincerely thank the editor and reviewers for their time and valuable comments, which have greatly helped us to improve the quality of our manuscript. We have made the following revisions to the abstract section.

Background: Depression is a prevalent psychological challenge during pregnancy, with established links to adverse outcomes like preterm birth (PTB) globally. However, epidemiological data from China's multiethnic regions are scarce, and experimental evidence supporting a causal relationship remains limited. This study aimed to investigate the association between prenatal depressive symptoms and the risk of PTB in a cohort from Yunnan, China, and to provide supportive evidence using a mouse model of depression.

Methods: We recruited 1,466 women during their first-trimester routine visits at Qujing Hospital. Depressive symptoms were assessed using the Chinese version of the Edinburgh Postnatal Depression Scale (EPDS), a screening tool, with a score ≥12 indicating elevated symptoms suggestive of depression. PTB was defined as delivery before 37 gestational weeks, confirmed by ultrasound. In parallel, a mouse model of depression was established using Chronic Unpredictable Mild Stress (CUMS) for 6 weeks prior to mating. PTB in mice was defined as delivery before 19 days of gestation.

Results:In the cohort study, the incidence of PTB was significantly higher in women with prenatal depressive symptoms compared to those without (8.43% vs. 3.83%, P < 0.001). The association remained significant after adjusting for sociodemographic and clinical confounders, with an adjusted risk ratio (aRR) of 2.19 (95% CI: 1.32-3.63). This association showed a significant dose-response pattern (P for trend = 0.03), with the risk being highest for women with moderate depressive symptoms (aRR = 2.44, 95% CI: 1.30-4.58). In the animal experiments, PTB did not occur in the control mice, whereas 40% of the mice exposed to CUMS (depression model group) delivered prematurely.

Conclusion: This study demonstrates a significant association between prenatal depressive symptoms and an increased risk of preterm birth in a Chinese multiethnic cohort. Experimental findings from a mouse model further suggest a potential contributory role of depression to PTB. These results underscore the importance of screening for and addressing maternal mental health during pregnancy. Future research should focus on underlying mechanisms and intervention strategies to mitigate this risk.

General comments:

Don’t state literature…in your text. Use previous studies

Introduction

What is meant by today? In the introduction add 2025 if it’s so. Along with months. I quickly checked the reference, but the reference is until 2021?

In the second para, spell what are the multiple risk factor?

In the same para you have described how it’s going to affect, add 1-2 lines as it’s the major objective of the study.

I am not clear by these statements what authors want to say.

“The strength of association between prenatal depression and PTB varied between ethnic groups, and the association between depression during pregnancy appears to be stronger in Asians Yunnan [25-27], as a multiethnic region, has not reported the association between depression during pregnancy and PTB, and it remains to be determined whether the effect of depression on PTB is different from that found in other regions”

Add rationale why do you think it varies with different ethnic origin.

The major missing is what is the rationale behind the use of animal models, add a paragraph explaining previous studies on animals’ models, what they found with respect to PTB and also report studies looking at antenatal depression on fetal brain development.

Add details of what is need for looking at a population from Yunnan. How does it vary from other Chinese province.

If you had any prior hypothesis add that too.

R We agree with the reviewer's suggestion for more precise language. We have replaced the term "literature" with "previous studies" or similar phrases throughout the manuscript to improve clarity and professionalism.

We thank the reviewer for pointing out this imprecision. We have removed the vague term "today" and rephrased the sentence to more accurately reflect the timeline of the existing evidence, as suggested.

This is a valuable suggestion to strengthen the logical flow of our introduction. We have now specified examples of risk factors and, as recommended, added a sentence to more explicitly connect the established knowledge about depression as a risk factor to the primary objective of our present study.

We appreciate the reviewer's feedback regarding the lack of clarity in this paragraph. We have thoroughly rewritten the problematic sentence to separate the distinct points and improve logical sequencing. Additionally, we have incorporated a rationale for why the association might vary by ethnicity, as requested, by mentioning potential contributing factors such as sociocultural and genetic influences. This revision, we believe, provides a much clearer and more compelling justification for conducting our study in Yunnan's multiethnic population.

We believe these additions have substantially enhanced the clarity, depth, and scholarly rigor of our introduction.

Methods:

I think you need to keep it as women with depressive symptoms, don’t just keep it as depressive alone it confuses.

Given the lack of my expertise in animal models I am not comments on animal model part, I have seen reviewers with that expertise have commented.

R We thank the reviewer for this critical comment. We agree that precise terminology is paramount. Throughout the manuscript, particularly in the Methods and Results sections, we have replaced potentially misleading terms such as "depressive" and "non-depressed" with more accurate descriptions based on the screening instrument used. The groups are now consistently referred to as "women with elevated depressive symptoms (EPDS ≥ 12)" and "women without elevated depressive symptoms (EPDS < 12)". We have also added explicit statements in the 'Screening Tools' subsection clarifying that the EPDS is a screening tool and not a diagnostic instrument. We believe these changes significantly improve the clarity and accuracy of our work.

We thank the reviewer for their time and acknowledge their expertise lies elsewhere. As the reviewer noted, other reviewers with specific expertise in animal models have provided their comments separately, and we have addressed those points in detail in other parts of our response letter.

Statistical analysis:

Can you add the assumption of Cox Proportion model.

Why was BMI, parity, marital status not considered? I would strongly suggest adding those in the model. For e.g., marital status I can see it in the Table 1. But still it play a major role.

I would suggest three models.

1. Crude model

2. Current model

3. Current model + BMI + Parity

Importantly, I am concerned that the recruitment happened during COVID 19 pandemic, how authors adjust control for it. See papers how they accounted for.

https://www.nature.com/articles/s41390-024-03620-7

https://www.sciencedirect.com/science/article/pii/S2949732924000735

Further in the screening tools for depression during pregnancy – when was the questionnaire administered. I can’t find any details of it.

How many responded and did not. Add those in Fig 1. Revise as depressive symptoms don’t keep it as diagnosis

R Response to Comments on Statistical Analysis and Methods:

Cox Model Assumption: We thank the reviewer for this suggestion. We have now explicitly stated that the proportional hazards assumption was tested using Schoenfeld residuals and was met, and we have included this in the revised Statistical Analysis section.

Model Specification and Additional Covariates: We agree with the reviewer that including pre-pregnancy BMI and parity strengthens the analysis. While these data were not initially available for all participants, we have since retrieved them from a subset of medical records for 89% of the cohort (n=1304). As recommended, we now present three models:

Model 1: no adjustment for confounding factors .

Model 2: adjust age, nationality, residence, marital status, education level, occupation status, annual family income, pregnancy.

Model 3 : adjust for all above + pre-pregnancy BMI and parity

The results, which remain robust across all models, are presented in the revised manuscript (Table 2, Page 14).

We sincerely thank the reviewer for raising this important point regarding the potential confounding effect of the COVID-19 pandemic. We fully acknowledge that the pandemic constituted a significant population-level stressor. We have carefully considered this issue and believe that our study design and complementary experimental evidence effectively address this concern, as detailed below.

1. Internal Comparison Within a Shared Context:

The core of our analytical approach is an internal comparison within a cohort that shared the same macro-environmental context of the pandemic. The pervasive stress of COVID-19, as the reviewer rightly notes, elevated the background level of anxiety and depressive symptoms for all participants in our study. However, our analysis does not compare a "pandemic group" to a "non-pandemic group"; rather, it compares women with high depressive symptoms (EPDS ≥12) to women with low depressive symptoms (EPDS <12) within the same pandemic context.

If the pandemic environment alone were the primary driver of preterm birth (PTB), we would expect to see little to no difference in PTB rates between these two groups, as both were exposed to the same overarching stressor. The fact that we observed a robust and significant increase in PTB risk specifically in the group with high depressive symptoms strongly suggests that the individual's psychological vulnerability and response—namely, the development of significant depressive symptoms—is the critical mediator of PTB risk, over and above the general background stress.

2. Corroborating Evidence from Animal Experiments:

Furthermore, our study is uniquely positioned to address this concern due to the inclusion of an animal model. The animal experiment was conducted in a controlled laboratory environment, entirely isolated from the human COVID-19 pandemic. In this setting, we induced depression-like behaviors in mice using the Chronic Unpredictable Mild Stress (CUMS) paradigm and observed a significantly higher rate of PTB compared to unstressed controls.

This finding is crucial because it demonstrates that a depression-like state can directly contribute to PTB in the absence of any pandemic-related factors. This experimental result provides independent, supportive evidence for a potential causal pathway from prenatal depression to PTB, which strengthens the interpretation of our human cohort findings.

3. Manuscript Revisions:

In direct response to this comment, we have revised the Discussion section of our manuscript to explicitly acknowledge the pandemic context and articulate the above arguments. We have added a new paragraph that clarifies the logic of our internal comparison and integrates the supporting evidence from our animal model .We believe that this combination of a robust study design and compelling complementary data provides a strong foundation for our conclusion that prenatal depressive symptoms are associated with an increased risk of PTB, independent of the shared context of the COVID-19 pandemic.

Fig 1 We have completed the revisions as requested.

Results:

“Notably, the incidence of depression in rural pregnant women was slightly higher than

that in urban pregnant women, with depression incidence rates of 26.01% and 18.77%,

respectively” Don’t compare it in results, just talk about your results, rest all move to discussion.

Same way remove this too

“which may reflect the narrow age

---

## [Decision Letter · Decision Letter 2]

21 Dec 2025

Dear Dr. Chen,

We look forward to receiving your revised manuscript.

Kind regards,

Samson Nivins, Ph.D

Academic Editor

PLOS One

Journal Requirements:

Additional Editor Comments:

All my comments were adequately addressed. Just do the corrections suggested by one of the reviewer.

Reviewers' comments:

Reviewer's Responses to Questions

**Comments to the Author**

Reviewer #5: All comments have been addressed

Reviewer #7: (No Response)

2. Is the manuscript technically sound, and do the data support the conclusions?

Reviewer #5: (No Response)

Reviewer #7: Yes

3. Has the statistical analysis been performed appropriately and rigorously?

Reviewer #5: (No Response)

Reviewer #7: Yes

4. Have the authors made all data underlying the findings in their manuscript fully available?

Reviewer #5: (No Response)

Reviewer #7: Yes

5. Is the manuscript presented in an intelligible fashion and written in standard English?

Reviewer #5: (No Response)

Reviewer #7: Yes

Reviewer #5: (No Response)

Reviewer #7: The authors have not adequately addressed this comment in the revised manuscript. Please provide a clear response and incorporate the necessary revisions. Language and Terminology:

The manuscript contains several grammatical and lexical errors that may hinder clarity.

Improving language clarity will enhance both readability and credibility.

Certain terms need to be revised for scientific accuracy. For example, the use of

“normal mice” to describe the control group is not recommended, as it can be

interpreted in a vague or colloquial manner. More appropriate terminology includes

“control mice” or “non-depressed mice,” which are standard in scientific literature and

precisely describe the group’s characteristics.

In the previous version of the manuscript, the following sentence appeared in the

Results section:

“Subgroup analysis revealed that pregnant women aged 25–34 years in the second

trimester with lower annual family income, higher education, living in urban areas,

having no occupation, and a married spouse were at higher risk for PTB due to

depression during pregnancy.”

Upon review, several linguistic and stylistic issues were identified that could affect

clarity and precision:

Lack of parallel structure:

The attributes in the list (e.g., with lower income, higher education, living in urban

areas, having no occupation) did not follow a consistent grammatical pattern. This was

revised to ensure structural uniformity and coherence.

Redundancy (“a married spouse”):

The phrase “a married spouse” was redundant, as the term “spouse” inherently implies

marriage. This might be corrected by using “married” to describe participant marital

status.

Stylistic improvement (“having no occupation”):

The phrase “having no occupation” might be replace with “unemployed”, which is more

concise and aligns with formal academic style.

Refined causal phrasing:

The expression “at higher risk for PTB due to depression during pregnancy” might be

modify to “at higher risk of PTB associated with depression during pregnancy”,

providing a more precise and neutral description of the relationship.

3. Animal Welfare Considerations:

In the section concerning animal welfare, it is important to emphasize the use of low-stress handling methods (e.g., using tunnels) outside of controlled procedures that may require a specific approach and tail handling. Please take this into consideration, as it may improve the quality of the results obtained.

4. Characterization of the Human Study Group:More detailed investigation of additional factors that may strongly influence depression, such as environmental conditions, social interactions, or genetic background, would help avoid false-positive or false-negative results. This is particularly important when the experimental group is small, as limited sample sizes can increase variability and reduce statistical power. Careful control and reporting of these factors would improve the reliability and interpretability of the findings.

**Do you want your identity to be public for this peer review?** For information about this choice, including consent withdrawal, please see our Privacy Policy

Reviewer #5: No

Reviewer #7: **Yes:** Katarzyna Kopaczka

---

## [Author Response · Author response to Decision Letter 3]

22 Dec 2025

1.The authors have not adequately addressed this comment in the revised manuscript. Please provide a clear response and incorporate the necessary revisions. Language and Terminology:

The manuscript contains several grammatical and lexical errors that may hinder clarity.

Improving language clarity will enhance both readability and credibility.

Certain terms need to be revised for scientific accuracy. For example, the use of

“normal mice” to describe the control group is not recommended, as it can be

interpreted in a vague or colloquial manner. More appropriate terminology includes

“control mice” or “non-depressed mice,” which are standard in scientific literature and

precisely describe the group’s characteristics.

R We sincerely thank the reviewer for pointing out the issues regarding language clarity and scientific terminology. We agree that precise language is crucial for the credibility and readability of the manuscript. We have thoroughly revised the manuscript to address these concerns:

Terminology for Animal Groups: As suggested, we have replaced the non-specific term “normal mice” throughout the manuscript with the more precise and standard terms “control mice” or “non-stressed control mice”. This change has been made in the Abstract, Results, and Discussion sections to accurately reflect the experimental design.

Language Clarity and Grammar: We have carefully proofread the entire manuscript to correct grammatical errors, improve sentence structure, and enhance overall clarity. Specific revisions include:

Correcting subject-verb agreement (e.g., “A questionnaire was employed”).

Refining awkward or imprecise phrases for better flow (e.g., rephrasing “comparison studies” to “A comparison of existing evidence”).

Replacing colloquial or inaccurate terms with scientific language (e.g., replacing “depressed patients” with “women with elevated depressive symptoms”).

Removing an out-of-context sentence in the Discussion that lacked prior methodological support.

We believe these comprehensive edits have significantly improved the linguistic quality and scientific rigor of the manuscript, addressing the reviewer’s valid concerns.

2.In the previous version of the manuscript, the following sentence appeared in the

Results section:

“Subgroup analysis revealed that pregnant women aged 25–34 years in the second

trimester with lower annual family income, higher education, living in urban areas,

having no occupation, and a married spouse were at higher risk for PTB due to

depression during pregnancy.”

Upon review, several linguistic and stylistic issues were identified that could affect

clarity and precision:

Lack of parallel structure:

The attributes in the list (e.g., with lower income, higher education, living in urban

areas, having no occupation) did not follow a consistent grammatical pattern. This was

revised to ensure structural uniformity and coherence.

Redundancy (“a married spouse”):

The phrase “a married spouse” was redundant, as the term “spouse” inherently implies

marriage. This might be corrected by using “married” to describe participant marital

status.

Stylistic improvement (“having no occupation”):

The phrase “having no occupation” might be replace with “unemployed”, which is more

concise and aligns with formal academic style.

Refined causal phrasing:

The expression “at higher risk for PTB due to depression during pregnancy” might be

modify to “at higher risk of PTB associated with depression during pregnancy”,

providing a more precise and neutral description of the relationship.

R We thank the reviewer for this meticulous and constructive feedback on the linguistic style of our results presentation. We fully agree that precision and clarity in wording are paramount.

We have revised the problematic sentence in the Results section to address all the raised points:

Parallel Structure: We have rephrased the list of characteristics to ensure grammatical consistency and structural coherence.

Redundancy & Style: We have eliminated the redundancy ("married spouse") and replaced the phrase "having no occupation" with the more concise and standard term "unemployed".

Causal Phrasing: We have replaced "due to" with "associated with" to more accurately describe the observed relationship, in line with the observational nature of our study.

The revised text now reads:

Pre-specified subgroup analyses were performed to assess the consistency of the association between depressive symptoms and PTB across different population strata (Figure 3). The increased risk of PTB associated with depressive symptoms was observed consistently across most subgroups. The point estimates were highest for women who were unemployed, resided in urban areas, or had a college/bachelor's degree or above.

We believe this revision significantly improves the clarity, precision, and academic tone of the manuscript.

3. Animal Welfare Considerations:

In the section concerning animal welfare, it is important to emphasize the use of low-stress handling methods (e.g., using tunnels) outside of controlled procedures that may require a specific approach and tail handling. Please take this into consideration, as it may improve the quality of the results obtained.

R We thank the reviewer for raising this important point regarding animal welfare and its impact on research quality. We fully agree that distinguishing between procedure-required handling and routine low-stress management is crucial.

To address this, we have added a dedicated subsection titled “Animal Handling and Welfare Considerations” within the Methods section (following the description of behavioral tests). In this new paragraph, we explicitly state:

That tail handling was limited to specific experimental protocols where it was an integral part of the test (e.g., the Tail Suspension Test).

That for all other routine husbandry and handling activities (e.g., transfers, identification), we prioritized the use of low-stress methods, such as handling tunnels or cupping, to minimize baseline stress.

This revision clarifies our comprehensive approach to animal welfare, aligns with best practices, and acknowledges the reviewer’s insight that such practices can contribute to the reliability of behavioral outcomes.

We believe this addition strengthens the methodological rigor of our manuscript.

4. Characterization of the Human Study Group:More detailed investigation of additional factors that may strongly influence depression, such as environmental conditions, social interactions, or genetic background, would help avoid false-positive or false-negative results. This is particularly important when the experimental group is small, as limited sample sizes can increase variability and reduce statistical power. Careful control and reporting of these factors would improve the reliability and interpretability of the findings.

R We sincerely thank the reviewer for raising this critical methodological point. We fully agree that unmeasured or insufficiently characterized confounding factors—such as detailed social environment, life history, and genetic background—can influence both depression risk and pregnancy outcomes, potentially affecting the validity of observed associations, especially in studies with moderate sample sizes.

In direct response to this comment, we have added a new, dedicated paragraph to the “Strengths and Limitations” section of the Discussion (now the second point). In this paragraph, we explicitly:

Acknowledge that our study did not capture key psychosocial and biological factors (e.g., social support, trauma history, genetic risk).

State that this represents a source of potential residual confounding.

Note the relevance of this limitation to our sample size considerations.

Propose that future research with larger cohorts incorporate more comprehensive assessments to address this gap.

We have also re-ordered the points in the Limitations section for better logical flow. We believe this addition significantly improves the transparency and intellectual honesty of our manuscript, providing readers with a clearer framework for interpreting our findings and identifying priorities for future research.

---

## [Editor Report · Decision Letter 3]

7 Jan 2026

Association between depression during pregnancy and preterm birth: results from population cohorts and mouse experimental models

PONE-D-24-60622R3

Dear Dr. Chen,

We’re pleased to inform you that your manuscript has been judged scientifically suitable for publication and will be formally accepted for publication once it meets all outstanding technical requirements.

Kind regards,

Samson Nivins, Ph.D

Academic Editor

PLOS One
---

## [Editor Report · Acceptance letter]

PONE-D-24-60622R3

PLOS One

Dear Dr. Chen,

I'm pleased to inform you that your manuscript has been deemed suitable for publication in PLOS One. Congratulations! Your manuscript is now being handed over to our production team.

Kind regards,

on behalf of

Dr. Samson Nivins

Academic Editor

PLOS One